# Mid-Holocene ITCZ migration: connection with Hadley cell dynamics and impacts on terrestrial hydroclimate

Jianpu Bian[1], Jouni Räisänen[1], and Heikki Seppä[2]

[1]Institute for Atmospheric and Earth System Research, University of Helsinki, Helsinki, Finland
[2]Department of Geosciences and Geography, University of Helsinki, Helsinki, Finland

**Correspondence:** Jianpu Bian (bian.jianpu@helsinki.fi)

**Abstract.** This study investigates the multiple changes of the Hadley cell (HC) in response to the northward migration of Intertropical Convergence Zone (ITCZ) and their combined influence on terrestrial hydrological cycle during the mid-Holocene, using simulations from the PMIP4-CMIP6 archive. Our results show that the annual global ITCZ position shifts northward by $0.2°$ and $0.3°$ as a multi-model mean using two different precipitation metrics, which is consistent with proxy evidence of a slight northward shift during the mid-Holocene. As the ITCZ is co-located with the rising branch of Hadley cell, the northward migration of ITCZ is accompanied by a northward movement of the inner HC edge. This further results in a contracted and weakened northern HC, while the southern HC expands and intensifies in the mid-Holocene. Specifically, the northern HC width contracted by $1.1°$ and $0.5°$, with strength reductions of $3.7\%$ and $4.1\%$, while the southern HC expanded by $1.2°$ and $0.6°$ and strengthened by $2.9\%$ and $1.8\%$, according to the two streamfunction metrics. Meanwhile, changes in stationary and transient eddies significantly influence the cross-equatorial energy transport during the mid-Holocene, complicating the link between energy transport and the ITCZ position. Moisture budget analysis shows that enhanced moisture eddy fluxes mainly contribute to increased terrestrial precipitation in the Northern Hemisphere, particularly in monsoonal regions, while Southern Hemisphere precipitation decreased due to evaporation and dynamic terms. Seasonal analysis indicates that the terrestrial hydrological cycle changes are primarily due to summer dynamics with an amplified inter-hemispheric contrast and asymmetry, while there are minor changes during winter seasons for both hemispheres. Although orbital forcing during the mid-Holocene was symmetric around the equator in the annual mean, it indirectly drove hemispherically asymmetric changes in annual atmospheric radiation balance. In the Northern Hemisphere, reduced surface shortwave radiation alongside increased atmospheric shortwave absorption indicates that enhanced cloudiness and water vapor play key roles. Moist static energy (MSE) budget analysis reveals that stronger rising motion significantly promotes vertical MSE advection over land in the Northern Hemisphere, enhancing moist convection and precipitation, while reduced rising motion weakens vertical MSE advection in the Southern Hemisphere, suppressing moist convection and precipitation over land. Regionally, ITCZ migration and associated HC changes alter climate patterns with reduced Northern Hemisphere terrestrial aridity and drylands contraction, while the Southern Hemisphere has enhanced aridity and drylands expansion. Multiple proxies support these findings, indicating wetter Northern Hemisphere conditions and a drier Southern Hemisphere, although inconsistencies remain in Australia's aridity pattern. Our results highlight the complex interactions among ITCZ migration, Hadley cell dynamics, global hydrological cycle, and terrestrial aridity during the mid-Holocene.

# 1 Introduction

During the mid-Holocene (MH; 6,000 years BP), currently arid parts of North Africa were transformed into thriving ecosystems with shrubs, grasslands, and forests (Claussen and Gayler, 1997; Claussen et al., 1999; Holmes, 2008; Harrison et al., 2014; Tierney et al., 2017; Pausata et al., 2020; Thompson et al., 2022; Kaufman and Broadman, 2023). For this warm and humid period characterized by significantly enhanced rainfall, numerous reconstruction and simulation based studies have sought to uncover the drivers behind this remarkable climatic shift (Claussen et al., 2017; Tierney et al., 2017; Pausata et al., 2020; Brierley et al., 2020; Thompson et al., 2022; Kaufman and Broadman, 2023).

The Hadley cell, characterized by rising air near the equator and descending air in the subtropics, plays a crucial role in regulating the Earth's energy budget by transporting energy from the tropics to higher latitudes (Diaz and Bradley, 2004). The ITCZ is a narrow equatorial band characterized by the convergence of trade winds, intense convective activity, dense cloud cover, and heavy precipitation. As a key component of large-scale atmospheric circulation, the ITCZ is intrinsically connected to the Hadley Cell, with their ascending branches tightly coupled in driving meridional and zonal overturning processes. The ITCZ and Hadley Cell play a pivotal role in shaping precipitation patterns and climate across the tropics and subtropics, significantly influencing monsoonal circulations and the seasonal migration of monsoons and rainfall (Diaz and Bradley, 2004). The positions of both the ITCZ and Hadley cell vary seasonally, following the Sun's migration and changes in Earth's perihelion precession, which have far-reaching effects on global hydrological cycle (Diaz and Bradley, 2004; Kang and Lu, 2012; Kang et al., 2018; Geen et al., 2020; Lionello et al., 2024).

During the mid-Holocene, changes in insolation influenced inter-hemispheric thermal contrast and enhanced seasonal variation in the cross-equatorial circulations (Diaz and Bradley, 2004; Harrison et al., 2014; Claussen et al., 2017; Brierley et al., 2020; Lionello et al., 2024), pushing the ITCZ northward beyond its present position (Schneider et al., 2014; Geen et al., 2020; Bian and Räisänen, 2024). Kang et al. (2008) demonstrated an anti-correlation between the annual ITCZ position and the cross-equatorial energy flux (EFE): the northward (southward) migration of the annual ITCZ position corresponds to an anomalous southward (northward) EFE. Further studies indicate that, in the annual mean, the global ITCZ position, the EFE, and the inner edge of global Hadley cell are nearly co-located (Donohoe et al., 2013; Donohoe and Voigt, 2017; Hill, 2019; Kang, 2020; Geen et al., 2020). This co-location implies that the ITCZ and the shared rising branch of the Hadley cell rely on atmospheric energy transport from the hemisphere in which the ITCZ is situated, with the energy export proportional to the ITCZ displacement away from the equator. Consequently, the ITCZ remains aligned with the ascending branch of the Hadley cell, meaning that changes in tropical precipitation with the ITCZ necessitate concurrent changes in Hadley cell (Donohoe and Voigt, 2017; Hill, 2019). Therefore, the ITCZ shift has a correlation with changes in the edges, width, and intensity of the Hadley cell in the mid-Holocene (Diaz and Bradley, 2004; Lau and Kim, 2015; Donohoe et al., 2013; Donohoe and Voigt, 2017; Byrne et al., 2018; Hill, 2019; Kang, 2020).

From an energetic constraint and eddy-mediated view, monsoons are parts of the large-scale tropical overturning circulations of the Hadley cell and ITCZ (Bordoni and Schneider, 2008; Schneider et al., 2014; Geen et al., 2020; Biasutti et al., 2018; Hill, 2019). The northward ITCZ shift and Hadley cell changes strengthen monsoonal circulations and drive a northward expansion

and significantly increased monsoonal rainfall during the mid-Holocene summer (Diaz and Bradley, 2004; Kang and Lu, 2012; Claussen et al., 2017; Tierney et al., 2017; D'Agostino et al., 2019; Geen et al., 2020; Kang, 2020; Lionello et al., 2024). Additionally, these shifts in the ITCZ and Hadley cell had profound effects on regional climates, altering aridity patterns and reshaping biome distributions across the tropics and subtropics (Diaz and Bradley, 2004; Lau and Kim, 2015). Understanding these changes in the Hadley cell and ITCZ is essential for assessing how large-scale circulation shifts drive transformations in the global hydrological cycle across different climate periods.

The energetic constraint theory offers a quantitative framework linking the annual ITCZ position and the Hadley cell, predicting that the annual ITCZ location is near the latitude ($\phi_{EFE}$) at which the vertically integrated meridional MSE flux within the Hadley cell approaches zero (Kang et al., 2008; Adam et al., 2016; Wei and Bordoni, 2018; Kang, 2020; Geen et al., 2020; Lionello et al., 2024). To simplify this complex relationship, classical energetic theory typically assumes that cross-equatorial atmospheric energy transport is dominated entirely by the Hadley cell, while transient and stationary eddy contributions are negligible (Byrne et al., 2018; Kang, 2020). However, this energetic theory has limitations on shorter time scales and regional ITCZ variations (Roberts et al., 2017; Kang, 2020), suggesting a potential gap between the seasonal evolution of $\phi_{EFE}$ and the ITCZ seasonal and regional migrations (Donohoe and Voigt, 2017; Wei and Bordoni, 2018; Kang et al., 2018; Kang, 2020; Geen et al., 2020). For this study, we primarily focus on annual changes in the global ITCZ position and its possible connection to Hadley cell changes between the PI and mid-Holocene.

Despite the many earlier studies, the complex interactions between the ITCZ, Hadley cell and hydrological changes during the mid-Holocene remain insufficiently understood. Therefore, a detailed quantitative evaluation of their dynamic interplay and multi-scale atmospheric processes involved is important, including changes in cross-equatorial energy flux transport by the stationary and transient eddies. This study addresses these gaps by focusing on three key aspects: (1) the dynamic connection between the northward ITCZ shift and Hadley cell changes during the mid-Holocene; (2) the joint influence of the ITCZ-Hadley cell evolution on the mid-Holocene hydrological cycle; and (3) the evaluation of consistency between proxy data and model simulations regarding terrestrial hydroclimate and land aridity in the mid-Holocene. Building on the recent advances in understanding the mid-Holocene global ITCZ by Bian and Räisänen (2024), this work specifically targets these critical research aspects.

Our primary focus for this study is to quantitatively access the connection between the global ITCZ northward migration and Hadley cell dynamics, and their combined impact on the global hydrological cycle and terrestrial aridity during the mid-Holocene. To achieve this, we employ various methods to analyze changes in the Hadley cell and ITCZ between the preindustrial and the mid-Holocene periods. Specifically, we use multiple mass stream-function metrics to assess the edges, width, and intensity of the Hadley cell, along with weighted precipitation metrics to determine the position of the ITCZ (Diaz and Bradley, 2004; Lau and Kim, 2015; Adam et al., 2016; Pikovnik et al., 2022; Lionello et al., 2024). Additionally, we compare our findings with proxy data and investigate the related physical mechanisms through moisture budget and moist static energy (MSE) budget analyses, exploring how these affect global hydrological cycle, terrestrial aridity, and dryland shifts during the mid-Holocene (Bartlein et al., 2011; Prado et al., 2013; Herbert and Harrison, 2016; Hill et al., 2017; Liu et al., 2019; Hill, 2019; Bian and Räisänen, 2024; Lowry and McGowan, 2024; Lionello et al., 2024).

## 2  Data and Methods

### 2.1  Data

We use monthly outputs of the MH simulations from the the Paleoclimate Modeling Intercomparison Project Phase 4 (PMIP4) (Otto-Bliesner et al., 2017) and the preindustrial (PI) Coupled Model Intercomparison Project Phase 6 (CMIP6) simulations (Eyring et al., 2016) from nine models (See further model details in Tables A1 and A2 of Appendix A). For this study, we analyze the last 100 years of simulations from each model. The simulations outputs are bilinearly interpolated to a grid resolution of $2.5° \times 2.5°$. For the evaluation of the ITCZ by precipitation metrics and Hadley cell by mass streamfunction metrics, these variables are further bilinearly interpolated into $0.5° \times 0.5°$ and $0.1° \times 0.1°$. We also tested conservative regridding of precipitation and mass streamfunction metrics for comparison, and our analysis indicates that differences in grid resolution and interpolation methods do not affect the results obviously. Additionally, we incorporate observations of present-day precipitation from the Global Precipitation Climatology Project (GPCP; 1979-2018; Adler et al. (2003)) monthly product to further evaluate the PMIP4-CMIP6 simulations results.

We further collect multiple types of reconstructions, and evaluate the PMIP4 simulations against global pollen-based proxies of mean annual precipitation (MAP) and the moisture availability (alpha) index from Bartlein et al. (2011), covering regions of Eurasia, Africa, Europe, and North America. For Australia, we supplement these reconstructions by merging data from the Indo-Pacific Pollen Database (Herbert and Harrison, 2016; Lowry and McGowan, 2024) with additional pollen records (See further details in Tables S1 and S2 of the Supplementary material). Due to the limited pollen records for South America, we further incorporate various types of proxies from Wirtz et al. (2010) and Prado et al. (2013). These proxies include oxygen and carbon isotopic fractionation, along with physical-chemical and biological indicators (Prado et al., 2013).

### 2.2  Defining metrics for the Hadley cell

We utilize two distinct meridional mass streamfunction ($\Psi(\phi, p)$) metrics to quantify the zonal and annual edges, as well as the width and strength of the Hadley cell (Diaz and Bradley, 2004; Nguyen et al., 2013; Pikovnik et al., 2022).

$$\Psi(\phi, p) = \frac{2\pi a \cos\phi}{g} \int_0^p [\overline{v}](\phi, p)\, dp \tag{1}$$

where $[\overline{v}](\phi, p)$ represents the zonal and time-averaged meridional wind at latitude $\phi$, and $p$ is air pressure.

The first Hadley cell edge metric employed is the mid-tropospheric mass streamfunction at 500 hPa (Kang and Lu, 2012; Pikovnik et al., 2022). The southern, inner and northern edge of the Hadley cell are defined as the three latitudes within $40°S$-$40°N$ where the zonal mean mass stream function crosses zero.

The mass streamfunction $\Psi(\phi, p)$ values are negative in the southern Hadley cell branch and positive in the northern Hadley cell branch, increasing poleward with latitude near the shared inner edge where the streamfunction is zero. We further use the

maxima (minima) of the meridional mass streamfunction at 500 hPa to quantify the strength of the Hadley cell ($\varpi_{HC}$) as

$$\begin{cases} \varpi_{NHC} = \Psi_{500hPa}|_{max} & \text{for} \quad \phi \in 10°S - 40°N \\ \varpi_{SHC} = \Psi_{500hPa}|_{min} & \text{for} \quad \phi \in 40°S - 10°N \end{cases} \tag{2}$$

where the streamfunction reaches its negative peak in the southern Hadley cell and its positive peak in the northern Hadley cell, respectively. The second Hadley cell edge and intensity metrics are defined analogously with Method I, but using the 900-to-200 hPa average of the streamfunction ($\langle\Psi\rangle_{900-200hPa}$) rather than the value at 500 hPa (Nguyen et al., 2013; Pikovnik

et al., 2022).

## 2.3 Defining ITCZ position metrics

We quantify the zonal and annual ITCZ location ($\phi_{ITCZ}$) by applying two different precipitation metrics (Adam et al., 2016). The first metric is the precipitation centroid for ITCZ location, denoted as

$$\phi_{ITCZ} = \frac{\int_{\phi_S}^{\phi_N} \phi [P(\phi)\cos\phi]^X \, d\phi}{\int_{\phi_S}^{\phi_N} [P(\phi)\cos\phi]^X \, d\phi} \tag{3}$$

where $P(\phi)$ is the zonal and time mean precipitation. The boundaries $\phi_S$ and $\phi_N$ are defined as $20°S/N$, and the integer power $X$ is 1 (Adam et al., 2016).

The second metric locates the ITCZ location where the total area-weighted precipitation between boundaries $\phi_S$ and $\phi_N$ are equally divided (Adam et al., 2016; Bian and Räisänen, 2024):

$$\int_{\phi_S}^{\phi_{ITCZ}} P(\phi)\cos\phi d\phi = \int_{\phi_{ITCZ}}^{\phi_N} P(\phi)\cos\phi d\phi \tag{4}$$

where $\phi_S$ and $\phi_N$ have the same definition as the first metric.

## 2.4 Column MSE budget analysis

Previous studies show that moist static energy (MSE) budget plays a critical role in shaping the terrestrial hydrological response to climate change (Hill et al., 2017; Hill, 2019; Geen et al., 2020; Lionello et al., 2024). Here we derive MSE budget terms from the column total energy budget equation (Neelin and Held, 1987; Chou et al., 2013; Byrne and Schneider, 2016; Hill

et al., 2017).

$$\frac{\partial A_E}{\partial t} + \nabla \cdot \mathbf{F}_a = R_a + LE + SH_0 \tag{5}$$

where $A_E$ is the column total atmospheric energy, consisting of column total potential energy, column kinetic energy, and column latent energy. $\mathbf{F}_a$ is the column total energy transport: $\langle(MSE + K)\mathbf{V}\rangle$, where $\langle\cdot\rangle$ indicates vertical integration over the mass of column. $MSE = C_pT + Lq + g \cdot z$, where $C_p$ denotes the specific heat at constant pressure, $g$ the gravitational

constant, $z$ the geopotential height, $L$ the latent heat of vaporization, and $q$ the specific humidity. $R_a$ represents the atmospheric

net radiation: $R_a = R_{TOA} - R_s$, where $R_{TOA}$ and $R_s$ are the net downward radiative fluxes at the top of the atmosphere (TOA) and the surface, respectively. Specifically, $R_{TOA} = SW_{\downarrow(TOA)} - SW_{\uparrow(TOA)} - LW_{\uparrow(TOA)}$, where $SW_{\downarrow(TOA)}$ and $SW_{\uparrow(TOA)}$ are the downward and upward shortwave radiation at the TOA, respectively, and $LW_{\uparrow(TOA)}$ are the upward longwave radiation at the TOA. $R_S = SW_{S\downarrow} - SW_{S\uparrow} + LW_{S\downarrow} - LW_{S\uparrow}$, where $SW_{S\downarrow}$ and $SW_{S\uparrow}$ are the upward and downward radiation at the surface, and $LW_{S\downarrow}$ and $LW_{S\uparrow}$ are the upward and downward radiation at the surface, respectively.

$LE$ and $SH_0$ denote the surface latent and sensible heat fluxes, respectively. Over a long-term time mean, the tendency term approaches zero, and the column kinetic energy divergence is much smaller compared to the column MSE flux divergence. The time-averaged column energy budget then becomes

$$\nabla \cdot \overline{\mathbf{F}_a} = \overline{R_a} + L\overline{E} + \overline{SH_0} \approx \nabla \cdot \overline{\langle (MSE)\mathbf{V}\rangle} \tag{6}$$

where the vertical energy flux difference between the top and bottom of the atmospheric column is balanced by divergence of MSE flux (Hill et al., 2017; Hill, 2019):

$$\nabla \cdot \overline{\langle (MSE)\mathbf{V}\rangle} = \langle \overline{\mathbf{V}} \cdot \nabla_p \overline{MSE}\rangle + \langle \overline{\omega} \cdot \frac{\partial}{\partial p}\overline{MSE}\rangle + \nabla \cdot \langle \overline{MSE'\mathbf{V}'}\rangle \tag{7}$$

where $\langle \overline{\mathbf{V}} \cdot \nabla_p \overline{MSE}\rangle$ denotes the column-integrated horizontal MSE advection under time mean, $\langle \overline{\omega} \cdot \frac{\partial}{\partial p}\overline{MSE}\rangle$ the column vertical MSE advection, and $\nabla \cdot \langle \overline{MSE'\mathbf{V}'}\rangle$ the column eddy MSE flux divergence.

For the change of $\nabla \cdot \overline{\mathbf{F}_a}$ from the PI to the MH period,

$$\delta(\nabla \cdot \overline{\mathbf{F}_a}) = \delta\overline{R_a} + L\delta\overline{E} + \delta(\overline{SH_0})$$
$$\approx \delta\langle \overline{\mathbf{V}} \cdot \nabla_p \overline{MSE}\rangle + \delta\langle \overline{\omega} \cdot \frac{\partial}{\partial p}\overline{MSE}\rangle + \delta\nabla \cdot \langle \overline{MSE'\mathbf{V}'}\rangle \tag{8}$$

where $\delta(\cdot)$ denotes $(\cdot)_{MH} - (\cdot)_{PI}$.

Eq. (8) can be further expanded as

$$\delta(\nabla \cdot \overline{\mathbf{F}_a}) \approx \underbrace{\langle \overline{\mathbf{V}} \cdot \nabla_p \delta(\overline{MSE})\rangle}_{I} + \underbrace{\langle \delta\overline{\mathbf{V}} \cdot \nabla_p \overline{MSE}\rangle}_{II} + \underbrace{\langle \delta\overline{\mathbf{V}} \cdot \nabla_p \delta(\overline{MSE})\rangle}_{non-linear}$$
$$+ \underbrace{\langle \overline{\omega} \cdot \frac{\partial}{\partial p}\delta(\overline{MSE})\rangle}_{III} + \underbrace{\langle \delta\overline{\omega} \cdot \frac{\partial}{\partial p}\overline{MSE}\rangle}_{IV} + \underbrace{\langle \delta\overline{\omega} \cdot \frac{\partial}{\partial p}\delta(\overline{MSE})\rangle}_{non-linear} \tag{9}$$
$$+ \underbrace{\delta(\nabla \cdot \langle \overline{MSE'\mathbf{V}'}\rangle)}_{V}$$

where the last term is evaluated as a residual:

$$\delta(\nabla \cdot \langle \overline{MSE'\mathbf{V}'}\rangle) \approx \delta\overline{R_a} + L\delta\overline{E} + \delta(\overline{SH_0}) - \delta\langle \overline{\mathbf{V}} \cdot \nabla_p \overline{MSE}\rangle - \delta\langle \overline{\omega} \cdot \frac{\partial}{\partial p}\overline{MSE}\rangle \tag{10}$$

As the two non-linear terms of Eq. (9) are minimal, we can approximately get the expression of $\delta(\nabla \cdot \overline{\mathbf{F}_a})$ as:

$$\delta(\nabla \cdot \overline{\mathbf{F}_a}) \approx I + II + III + IV + V \tag{11}$$

Thus, changes in the MSE budget have five contributing terms in Eq. (11): anomalous column-integrated horizontal advection (thermodynamic term I, and dynamic term II), anomalous column-integrated vertical advection (thermodynamic term III, and dynamic term IV), and anomalous column-integrated eddy MSE flux divergence term V.

The five RHS terms play a critical role in influencing moist convection and precipitation patterns (Hill et al., 2017; Hill, 2019; Lionello et al., 2024). During the mid-Holocene, increased solar radiation modified the meridional thermal contrast in the Northern Hemisphere, altering the MSE gradient and advection, and further enhancing moist convection and precipitation. Horizontal MSE advection is relatively weak in the tropics compared to vertical advection, which has a closer connection with atmospheric energy flux divergence and vertical motion. Vertical MSE advection is primarily driven by the time-mean divergent circulation as a balancing mechanism and is sensitive to the depth of moist convection, which can significantly influence atmospheric instability, the development of convective systems, and changes in precipitation (Back and Bretherton, 2006; Chou et al., 2013; Hill, 2019; Geen et al., 2020; Lionello et al., 2024).

## 3 Results

### 3.1 Annual hydrological changes in simulations and reconstructions

During the PI period, the tropics exhibits intense rainfall ($\geq$ 3 mm/day), concentrated primarily between $20°S$ and $20°N$ (Figure 1a), which closely aligns with the observed present-day precipitation patterns from the GPCP dataset (not shown). As illustrated in the Taylor diagram (Taylor (2001); Figure 1c), the PI simulations effectively reproduce the large-scale annual precipitation distribution, showing minimal biases in magnitude and spatial variance when compared to GPCP observations. Notably, the multimodel means outperform most individual models, making them the preferred choice for subsequent analysis. In the zonal-mean framework, the deep tropics are characterized by intense precipitation and strong rising motion, with two distinct peaks centered around $10°S$ and $10°N$ (Figure 1d). During the mid-Holocene, annual precipitation increases significantly in the monsoonal regions and marine subtropics of the Northern Hemisphere (Figure 1b). Additionally, regional marine precipitation increases in the South Pacific and western Indian Ocean, whereas most other tropical ocean regions experience reduced rainfall in the mid-Holocene.

As horizontal advection is relatively weak in the low latitudes, there is a connection between atmospheric energy flux divergence and rising motion by $\nabla \cdot \overline{\mathbf{F}_a} > 0$. Due to the upward increase in total energy content, positive energy flux divergence requires upper tropospheric divergence and lower tropospheric convergence, which necessitates mid-tropospheric rising motion. Consequently, increases in precipitation ($\delta P > 0$) are generally associated with an increase in atmospheric energy flux divergence ($\delta \nabla \cdot \overline{\mathbf{F}_a} > 0$), and decreases in $\nabla \cdot \overline{\mathbf{F}_a}$ are usually associated with a decrease in precipitation ($\delta P < 0$). There is a good agreement between the changes in annual precipitation (Figure 1b) and those in the energy flux divergence ($\delta \nabla \cdot \overline{\mathbf{F}_a}$ in Figure 1e).

The mid-Holocene pollen reconstructions presented by Bartlein et al. (2011) indicate wetter conditions in the Northern Hemisphere (Figure 1f), with increased annual precipitation across tropical and subtropical regions in northern Africa, Europe, East Asia, western and southern North America, which align with the results of the PMIP4 simulations shown in Figure 1b.

In contrast, equatorial regions including tropical Africa, experienced reduced rainfall. For the Southern Hemisphere, pollen records indicate wetter conditions in southwestern Africa but a markedly drier climate in southeastern Africa during the mid-Holocene (Figure 1f).

In South America, multiple types of proxies generally suggest drier conditions and reduced rainfall across the continent during this period (Wirtz et al., 2010; Prado et al., 2013), as shown in Figures 1b and 1g. A few sites in northeastern Brazil show evidence of wetter conditions (Figure 1g), likely due to enhanced regional land–sea breeze under the changes in the position and intensity of the South Atlantic subtropical high during the mid-Holocene (Nagai et al., 2009; Prado et al., 2013). In Australia, however, the known problem of spatial discrepancies between simulated precipitation changes and reconstructions still remains (Krause et al., 2019; Liu et al., 2019; D'Agostino et al., 2020; Lowry and McGowan, 2024), as shown in Figures 1b and 1f. For northern Australia, recent studies indicate that the contraction of the local ITCZ contributed to reduced monsoon activity and precipitation (Reeves et al., 2013; Proske et al., 2014; Field et al., 2017; Lowry and McGowan, 2024), which aligns with PMIP4 simulation results (Figures 1b and 1f). For temperate Australia, some sites of pollen reconstructions and paleo-hydrologic records of the Oz-INTIMATE series in southeastern Australia indicate wetter conditions with enhanced river discharge and increased precipitation during the mid-Holocene (Petherick et al., 2013; Herbert and Harrison, 2016; Lowry and McGowan, 2024), while PMIP4 simulations do not consistently capture those robust changes in Figure 1b.

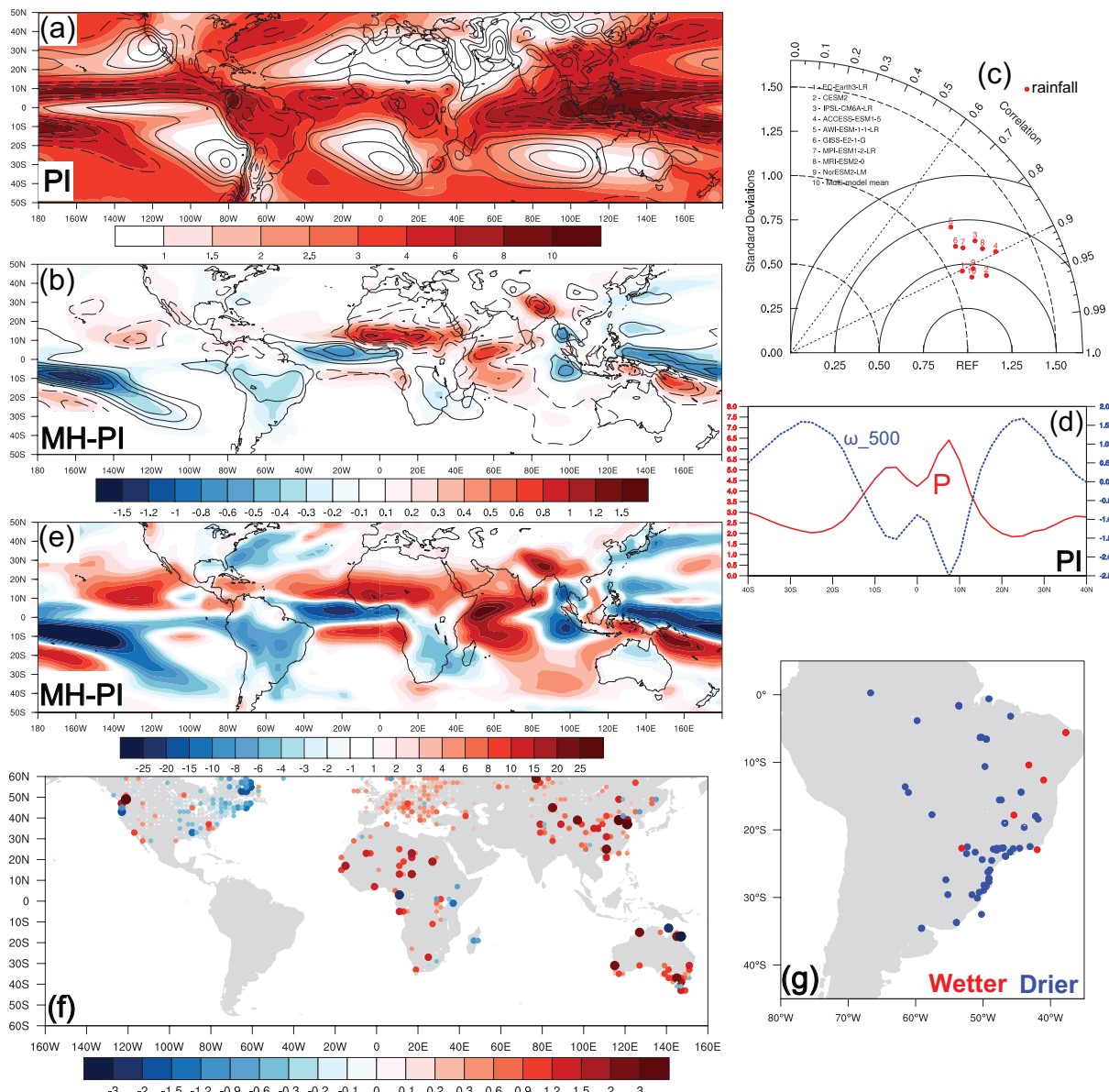

**Figure 1.** (a) PMIP4-CMIP6 multimodel annual mean precipitation (colored; unit: mm/day) and $\omega$ at 500 hPa (contour) in the PI period. (b) as (a), but for the changes of annual precipitation and $\omega$ at 500 hPa from the PI to the MH period. The solid (dashed) contours indicate positive (negative) anomalies of $\omega$ at 500 hPa, with an interval of $10^{-2}$ and $0.2 \cdot 10^{-2} \, Pa \, s^{-1}$ in (a) and (b), respectively. (c) the Taylor diagram for the global distribution of annual mean rainfall between each model in the PI simulation and the observation (GPCP (Adler et al. (2003)), 1979-2018, as the REF). (d) Annual and zonal mean precipitation and $\omega$ at 500 hPa during the PI period. (e) the changes of the divergence of column total energy transport (unit: $Wm^{-2}$). See Eq. (9) for further details. (f) Reconstructed annual precipitation changes (unit: mm/day) from pollen proxies (Bartlein et al., 2011; Herbert and Harrison, 2016; Lowry and McGowan, 2024). Sites with red color indicate increased mid-Holocene rainfall compared to present-day (0 ka) proxy records, and blue ones for reduced rainfall. Note that large dots represent significant changes of reconstructed annual precipitation (exceeding two standard errors), and small ones are not significant, as mentioned in Bartlein et al. (2011). (g) Multiple types of proxies from other compilations in the South America (Wirtz et al., 2010; Prado et al., 2013).

**Table 1.** Changes in the width ($^\circ N$) and strength (%) of the Southern and Northern Hadley cells from the PI to the MH period ($\Psi_{500hPa}$ for Method I; $\langle\Psi\rangle_{900-200hPa}$ for Method II)

| Model | ΔWidth_S | | ΔWidth_N | | ΔStrength_S | | ΔStrength_N | |
|---|---|---|---|---|---|---|---|---|
| | **I** | **II** | **I** | **II** | **I** | **II** | **I** | **II** |
| 1 | 2.3 | 1.0 | -2.3 | -1.0 | 6.6% | 4.5% | -6.9% | -6.6% |
| 2 | -0.2 | 0.5 | 0.0 | -0.3 | 2.1% | 1.2% | -0.1% | -1.4% |
| 3 | 0.8 | 0.3 | -0.5 | -0.3 | 3.3% | 1.7% | -5.2% | -5.2% |
| 4 | 1.3 | 0.5 | -1.5 | -0.5 | 4.0% | 3.3% | -3.1% | -3.6% |
| 5 | 1.3 | 0.5 | -0.8 | 0.0 | 1.2% | 0.6% | -4.9% | -6.0% |
| 6 | 0.5 | 0.3 | -0.3 | -0.3 | 1.1% | -0.2% | -4.2% | -3.9% |
| 7 | 2.8 | 1.3 | -2.0 | -0.5 | 1.2% | 0.03% | -5.5% | -5.8% |
| 8 | 1.3 | 0.8 | -1.3 | -1.0 | 2.9% | 1.6% | -2.2% | -3.6% |
| 9 | 0.5 | 0.5 | -1.0 | -0.5 | 3.9% | 3.2% | -0.8% | -1.2% |
| **Mean** | **1.2** | **0.6** | **-1.1** | **-0.5** | **2.9%** | **1.8%** | **-3.7%** | **-4.1%** |

## 3.2 Hydrological effects of the ITCZ shift and associated Hadley cell changes

To examine the relationship between changes in the ITCZ and the Hadley cell, we use two different precipitation metrics to quantify the ITCZ location, one defined as the precipitation centroid and the other as the median latitude of precipitation between $20^\circ S$ and $20^\circ N$ (See Eqs.(3)-(4) for further details), and the two different streamfunction metrics defined in Section 2.2 to measure the edges, widths, and strengths of the northern and southern Hadley cells.

As shown in Table 1, the changes in width and strength are positively correlated among the models. A wider Hadley cell tends to accommodate greater total upward and downward mass fluxes, and is therefore stronger as measured by the streamfunction maximum, even if the intensity of vertical motions remains relatively stable. Compared to the PI period (Figure 2a), the northern Hadley cell becomes narrower with weaker overturning circulation, which is associated with northward shifted rising motion and reduced subsidence in the northern part of the cell as shown in Figure 2b. Specifically, the ensemble mean PMIP4 simulations indicate that the strength of the northern Hadley cell during the mid-Holocene is reduced by 3.7% and 4.1% according to the two different streamfunction metrics (Table 1). For changes in the northern Hadley cell width, the ensemble mean result shows a contraction by $1.1^\circ$ and $0.5^\circ$, respectively.

Conversely, the expansion of the southern Hadley cell, as indicated by the two mass streamfunction metrics in Table 1, is accompanied by enhanced strength. This expansion suggests a stronger and northward extended overturning circulation with widespread tropospheric drying south of the equator (Figures 2b and 2d). During the mid-Holocene, the southern Hadley cell strength by ensemble mean is enhanced by 2.9% and 1.8% using the two metrics (Table 1). Additionally, the ensemble mean changes in the southern Hadley cell width show and expansion of $1.2^\circ$ and $0.6^\circ$ in latitude, respectively.

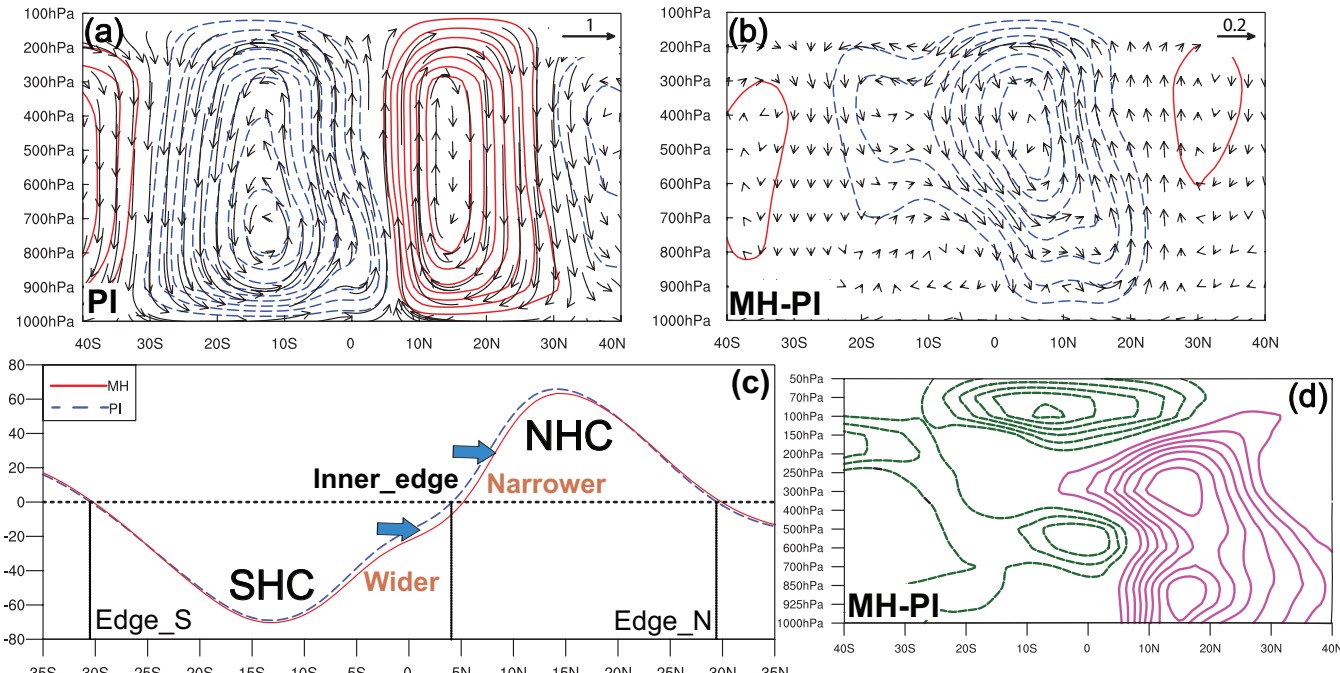

**Figure 2.** (a) Annual mean Hadley cell during the PI period, with solid red contours representing positive values and dashed blue contours indicating negative values of the meridional mass streamfunction (unit: $10^{10}$ kg/s). Meridional circulation ($v$, - $\omega$) is shown by vectors, scaled by $1\ ms^{-1}$ and $10^{-2}\ Pa\ s^{-1}$, respectively. (b) Changes of meridional mass streamfunction in the mid-Holocene. The streamfunction contours have an interval of $10^9$ kg/s, and the meridional circulation changes are scaled by $0.2\ ms^{-1}$ and $0.2 \cdot 10^{-2}\ Pa\ s^{-1}$, respectively. (c) Meridional Hadley cell depicted by the zonal mean, vertically-weighted mass streamfunction between 900 and 200 hPa during the PI (blue) and the MH (red) periods (unit: $10^9$ kg/s). Vertical lines in (c) indicate the northern edge (Edge_N), inner edge, and southern edge (Edge_S) of Hadley cell. (d) as (b), but for changes of zonal mean relative humidity. Solid (dashed) contours indicate positive (negative) anomalies (interval: 0.5%). All results are based on PMIP4-CMIP6 multimodel averages.

Our findings further suggest that these changes in the widths of the southern and northern Hadley cells are primarily driven by a northward shift of the inner edge in the mid-Holocene, as illustrated in Figure 2c. The mid-tropospheric streamfunction

metric ($\Psi_{500hPa}$) reveals an ensemble mean northward shift of 1.4° for the Hadley cell inner edge during the mid-Holocene, which is larger than the shift based on the vertically weighted streamfunction metric ($\langle\Psi\rangle_{900-200hPa}$) by 0.7° (Figures 3a, 3b). The intermodel standard deviations (STD) for the change in the Hadley cell inner edge shift is 0.8° for the $\Psi_{500hPa}$ metric, and 0.3° for the $\langle\Psi\rangle_{900-200hPa}$ metric. Both STDs are smaller than the multi-model mean changes, with closer agreement observed for the Hadley cell inner edge changes when using the $\langle\Psi\rangle_{900-200hPa}$ metric (Figures 3a and 3b).

The mid-Holocene shifts in the global and annual ITCZ positions, derived from the first and second precipitation metrics, reveal an average northward migration of 0.2° and 0.3°, respectively (Figures 3c and 3d). These findings align closely with multiple mid-Holocene proxy records, which suggest a northward ITCZ migration of the ITCZ up to 1° (Haug et al., 2001;

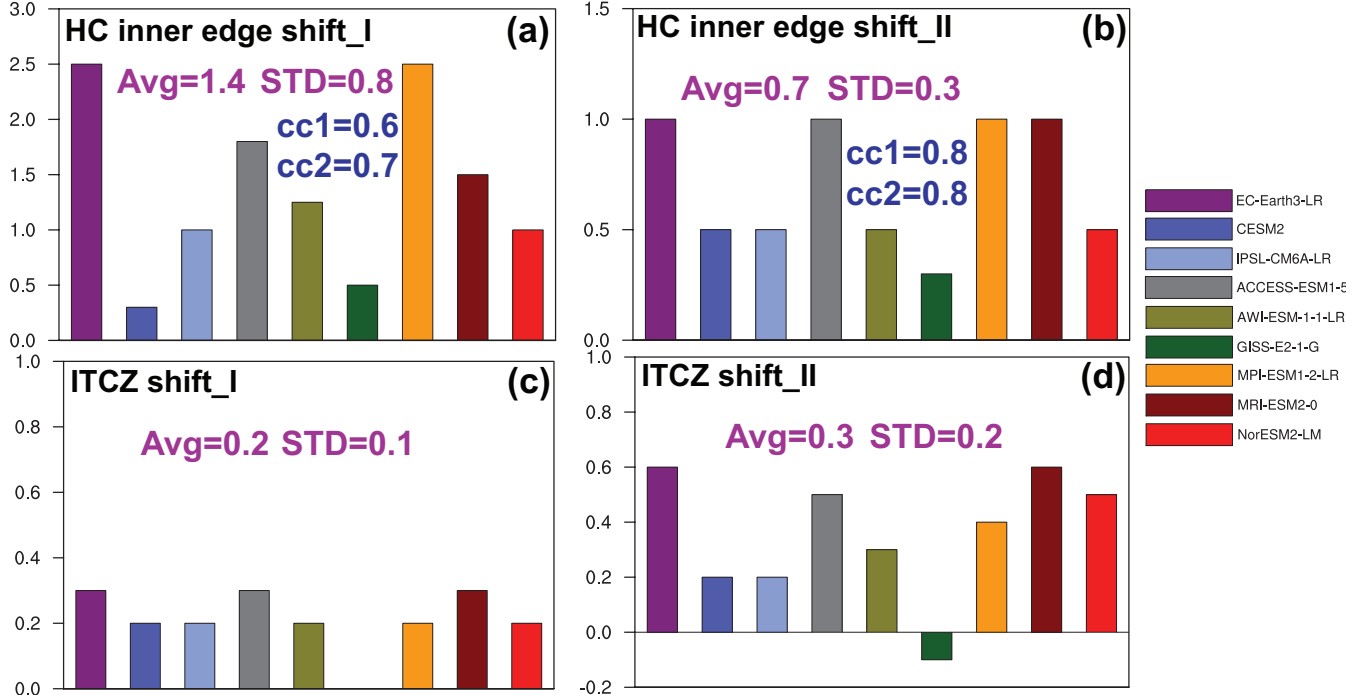

**Figure 3.** (a) and (b) are the latitude changes ($^\circ N$) of Hadley cell inner edge using the two streamfunction metrics, respectively. (c) and (d) are the latitude changes ($^\circ N$) of ITCZ position using the two precipitation metrics, respectively. Note the change of GISS-E2-1-G in (c) is zero. $cc1$ and $cc2$ in (a) denote the intermodel correlation coefficients between the shifts in the Hadley cell inner edge using the first streamfunction metric and the shifts in the ITCZ position by the two precipitation metrics, respectively. $cc1$ and $cc2$ in (b) are as in (a), but for the shifts of Hadley cell inner edge using the second streamfunction metric.

McGee et al., 2014). Furthermore, the STDs of the ITCZ location change by the first and second metrics are $0.1^\circ$ and $0.2^\circ$, respectively, as shown in Figures 3c and 3d. The first (precipitation centroid) metric thus exhibits a better agreement among

models with a smaller STD.

Previous studies of global warming based on observations and climate simulations reveal that the deep-tropics squeeze and the projected northward migration of ITCZ tend to influence the changes in the width and strength of tropical overturning circulation (Hadley cell) (Kang and Lu, 2012; Lau and Kim, 2015; Byrne et al., 2018; Watt-Meyer and Frierson, 2019; Lionello et al., 2024). In particular, the changes of annual ITCZ position co-vary with the inner edge changes of Hadley cell as well as the

annual cross-equatorial atmospheric energy flux (Donohoe and Voigt, 2017; Hill, 2019). Therefore, changes of inner edge could further affect the Hadley cell width and strength. During the mid-Holocene, our results indicate that the northward movement of inner edge contributes to changes in the Hadley cell width, while its outer edges have minor changes. Figures 3a and 3b show a strong positive correlation between shifts in the global ITCZ location and the inner boundary of the Hadley cell during the mid-Holocene across all the metrics used in this study. For the $\Psi_{500hPa}$ metric to define the Hadley cell inner edge shift,

its correlation with the two precipitation-based metrics of ITCZ location shift is 0.6 and 0.7 (Figure 3a). The corresponding

correlation when basing the Hadley cell shift on the $\langle\Psi\rangle_{900-200hPa}$ metric is slightly higher at 0.8 for both two precipitation metrics (Figure 3b).

The changes between the ITCZ and the Hadley cell were closely linked during the mid-Holocene as discussed above, while their relationship is inherently nonlinear and complex (Watt-Meyer and Frierson, 2019). Although they share the ascending branch near the Equator and are correlated via the cross-equatorial atmospheric energy flux under annual-mean state (Watt-Meyer and Frierson, 2019; Hill, 2019), a complete understanding of the factors driving their changes may be difficult to achieve when considering the influence from the extratropics and multiple climate drivers (Byrne et al., 2018; Kang et al., 2018; Kang, 2020; Geen et al., 2020; Lionello et al., 2024).

To simplify this complexity, the classical energetic theory for the ITCZ position assumes that cross-equatorial atmospheric energy flux arises entirely from the Hadley cell, with negligible contributions from transient and stationary eddies (Byrne et al., 2018; Kang, 2020). Within this framework, the ITCZ position is predicted by the latitude where the vertically integrated meridional MSE flux crosses zero (Kang et al., 2008; Adam et al., 2016; Donohoe and Voigt, 2017; Wei and Bordoni, 2018; Kang, 2020). As shown in Figure B1, the annual cross-equatorial energy flux (Eq. B1) exhibits negative anomalies ($\delta\overline{F} < 0$) in the deep tropics as well as the Northern Hemisphere, indicating reduced northward atmospheric energy transport from the Southern Hemisphere during the mid-Holocene. This imbalance shifts the energy flux zero-crossing latitude northward, resulting in a corresponding northward migration of the ITCZ. However, as shown in Figure B1, the contributions from the stationary and transient eddies changes (Eqs. B2-B3) are not negligible during this period. These complexities highlight the limitations of the energetic constraint framework in fully explaining ITCZ shifts and their nonlinear relationship with Hadley cell dynamics during the mid-Holocene.

Understanding changes in the Hadley cell and ITCZ is crucial for comprehending how alterations in large-scale circulations further affect the global hydrological cycle during the mid-Holocene. To investigate this, we conduct the moisture budget analysis by using mass-weighted vertical integration

$$\delta\overline{P} = \delta\overline{E} \underbrace{-\langle\overline{\mathbf{V}}\cdot\nabla_p\delta\overline{q}\rangle - \langle\delta\overline{q}\nabla_p\cdot\overline{\mathbf{V}}\rangle}_{\delta TH} \underbrace{-\langle\delta\overline{\mathbf{V}}\cdot\nabla_p\overline{q}\rangle - \langle\overline{q}\nabla_p\cdot\delta\overline{\mathbf{V}}\rangle}_{\delta DY} \underbrace{-\langle\nabla_p\cdot\delta(\overline{\mathbf{V}'q'})\rangle}_{\delta TE} + \delta S \tag{12}$$

where $\delta(\cdot)$ denotes $(\cdot)_{MH} - (\cdot)_{PI}$. $\overline{P}$ is the annual mean precipitation, $\overline{q}$ the annual mean specific humidity, $\overline{E}$ the annual evaporation, and $\mathbf{V}$ the annual mean horizontal wind. The vertical integration $\langle\cdot\rangle$ is from 1000 to 100 hPa. Eq. (12) further decomposes the precipitation change to contributions from five RHS terms, including the evaporation term ($\delta\overline{E}$), the thermodynamic term ($\delta TH$), the dynamic term ($\delta DY$), the transient eddy flux term ($\delta TE$), and the surface term ($\delta S$). Furthermore, $\delta S$ and $\delta TE$ are approximately derived as: $\delta S \approx -\delta(\nabla_p\cdot\langle\overline{\mathbf{V}q}\rangle) - \delta TH - \delta DY$, and $\delta TE \approx \delta\overline{P} - \delta\overline{E} + \delta(\nabla_p\cdot\langle\overline{\mathbf{V}q}\rangle)$, following Seager and Henderson (2013). The transient eddy contribution ($\delta S$) that requires daily data for direct calculation can be approximately evaluated as a residual of the time mean moisture budget.

During the mid-Holocene, there is a reduction in moisture convergence in the tropics, as shown by the negative $\overline{P} - \overline{E}$ anomalies in Figure 4a, while the subtropics show a wetting trend, particularly in the Northern Hemisphere's subtropical regions, accompanied by positive $\overline{P} - \overline{E}$ anomalies. The comparison between continents and oceans shows that the northward shift in annual rainfall associated with the ITCZ and Hadley cell predominantly occurs over land (Bian and Räisänen, 2024),

as shown in Figures 4c and 4e. Specifically, the transient eddy flux term ($\delta TE$) in Eq. (12) emerges as the leading contribution to the increase in terrestrial precipitation in the Northern Hemisphere (Bian and Räisänen, 2024), with evaporation playing a secondary role, and dynamic ($\delta DY$) and thermodynamic ($\delta TH$) terms have a comparatively smaller influence (Figure 4c). Additionally, for the reduced precipitation in the Southern Hemisphere, the evaporation and dynamic terms play a primary role.

In summary, the northward migration of annual ITCZ and shared rising branch is accompanied by a contracted and weakened northern Hadley cell in the mid-Holocene. The weaker overturning circulation and reduced subtropical descending branch of northern Hadley cell lead to stronger rising motion, wetter tropospheric conditions, and increased terrestrial precipitation in the Northern Hemisphere. By contrast, the southern Hadley cell expands and intensifies during this period, accompanied by enhanced overturning circulation and strengthened subtropical descending branch, leading to drier tropospheric conditions, and decreased terrestrial precipitation within the southern Hadley cell region. Further moisture budget analysis reveals that the eddy flux term ($\delta TE$) primarily contributes to increased terrestrial precipitation in the Northern Hemisphere. Conversely, terrestrial precipitation reduction in the Southern Hemisphere is primarily due to evaporation and dynamic terms.

## 3.3 Physical mechanisms driving terrestrial hydrological cycle changes

The changes in perihelion precession and altered insolation patterns in the mid-Holocene influenced seasonal and inter-hemispheric thermal contrast between winter and summer (Harrison et al., 2014; Claussen et al., 2017; Brierley et al., 2020), intensifying the seasonal variation in the cross-equatorial Hadley cell and ITCZ (Diaz and Bradley, 2004; Harrison et al., 2014; Claussen et al., 2017; Brierley et al., 2020; Lionello et al., 2024). From an energetic constraint and eddy-mediated view, monsoons as integral components of the large-scale tropical overturning circulations of the Hadley cell and ITCZ (Bordoni and Schneider, 2008; Schneider et al., 2014; Geen et al., 2020; Biasutti et al., 2018; Hill, 2019), further exhibited pronounced seasonal migration and contrast with significantly modulated monsoonal rainfall patterns during the mid-Holocene (Claussen et al., 2017; Tierney et al., 2017; D'Agostino et al., 2019; D'Agostino et al., 2020; Geen et al., 2020; Kang, 2020; Lionello et al., 2024). Therefore, the northward ITCZ migration and associated Hadley cell changes can further lead to shifts in terrestrial hydroclimates (Diaz and Bradley, 2004; Hill et al., 2017; Hill, 2019; Lionello et al., 2024). To explore these changes, we further investigate the seasonal evolution of terrestrial hydrological cycle and energy constraints influencing the terrestrial hydroclimate during this period.

Consistent with the analysis in Section 3.2, Figure 5 further illustrates that the seasonal evolution in the hydrological cycle exhibits contrasting patterns between terrestrial and marine areas. Over land, key terrestrial hydrological components, including precipitation (Figure 5a), surface evaporation (Figure 5c), runoff (Figure 5e), and mid-tropospheric vertical motion (Figure 5g), show significant increases in the Northern Hemisphere during the mid-Holocene summer season (June-to-September; JJAS). This intensification is particularly evident in the West African and Asian Monsoon systems in the Northern Hemisphere, where monsoonal rainfall is greatly enhanced (Pausata et al., 2016; Claussen et al., 2017; D'Agostino et al., 2019; Brierley et al., 2020; Bian and Räisänen, 2024). In contrast, the Southern Hemisphere summer season (December-to-March; DJFM) exhibits an opposite change of terrestrial hydrological cycle compared to the Northern Hemisphere, which is largely attributed to the suppression of subtropical monsoonal systems and reduced monsoonal precipitation (D'Agostino et al., 2020; Bian

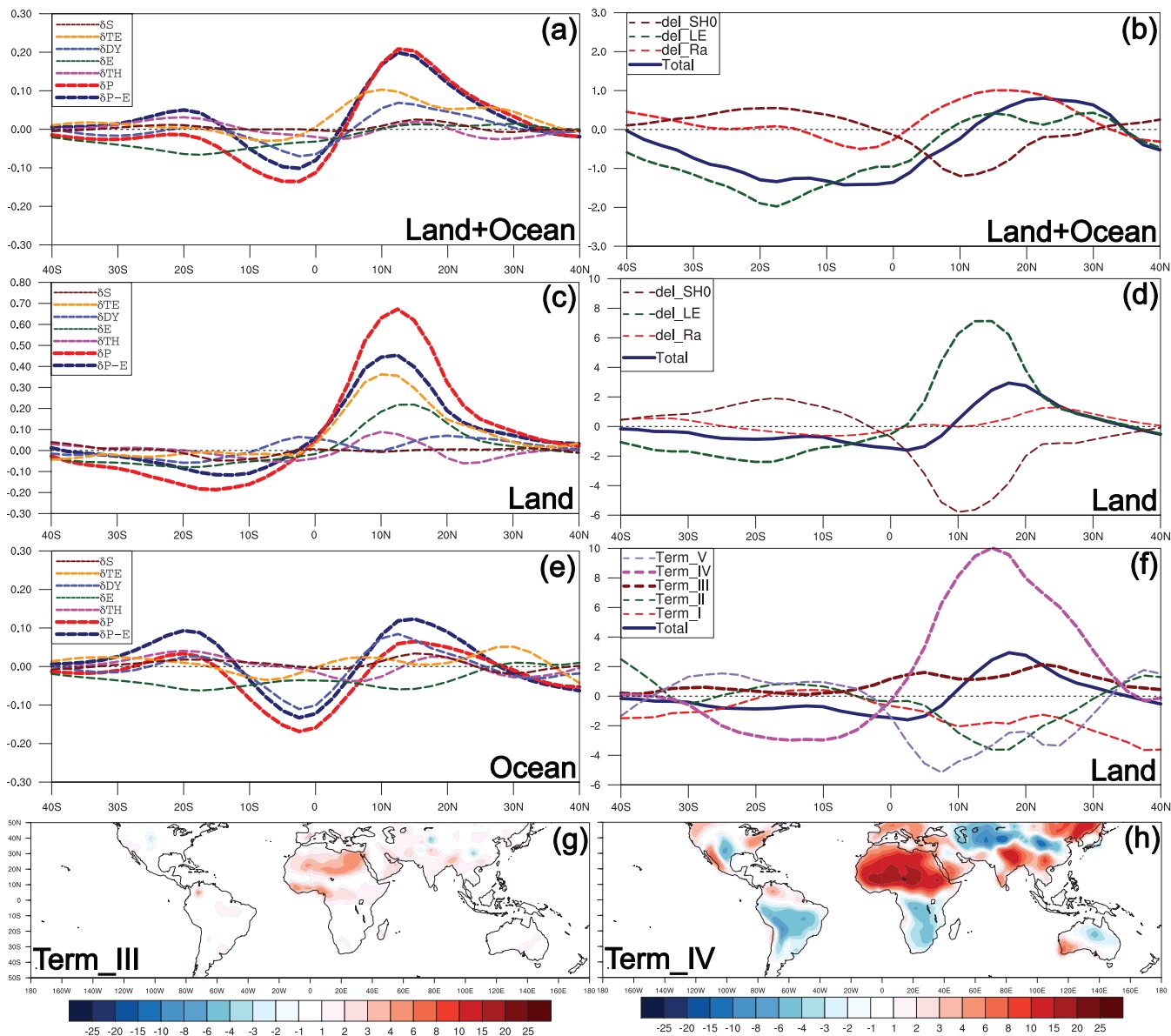

**Figure 4.** (a) PMIP4-CMIP6 multimodel annual and zonal mean changes of moisture budget (Eq. (12); unit: mm/day). (c) and (e) are as (a), but for the changes in the moisture budget over land and ocean, respectively (unit: mm/day). (b) PMIP4-CMIP6 multimodel annual and zonal mean changes of atmospheric energy source terms from the PI to the MH period (unit: $Wm^{-2}$). (d), (f) are as (b), but for changes of column energy balance and associated RHS terms over land (unit: $Wm^{-2}$), respectively. See Eqs. (7)-(9) for further details. (g) and (h) show the distribution of the RHS terms III and IV in Eq. (9) over land, respectively.

and Räisänen, 2024). Meanwhile, for winter seasons, both hemispheres experience minor alterations over land, emphasizing

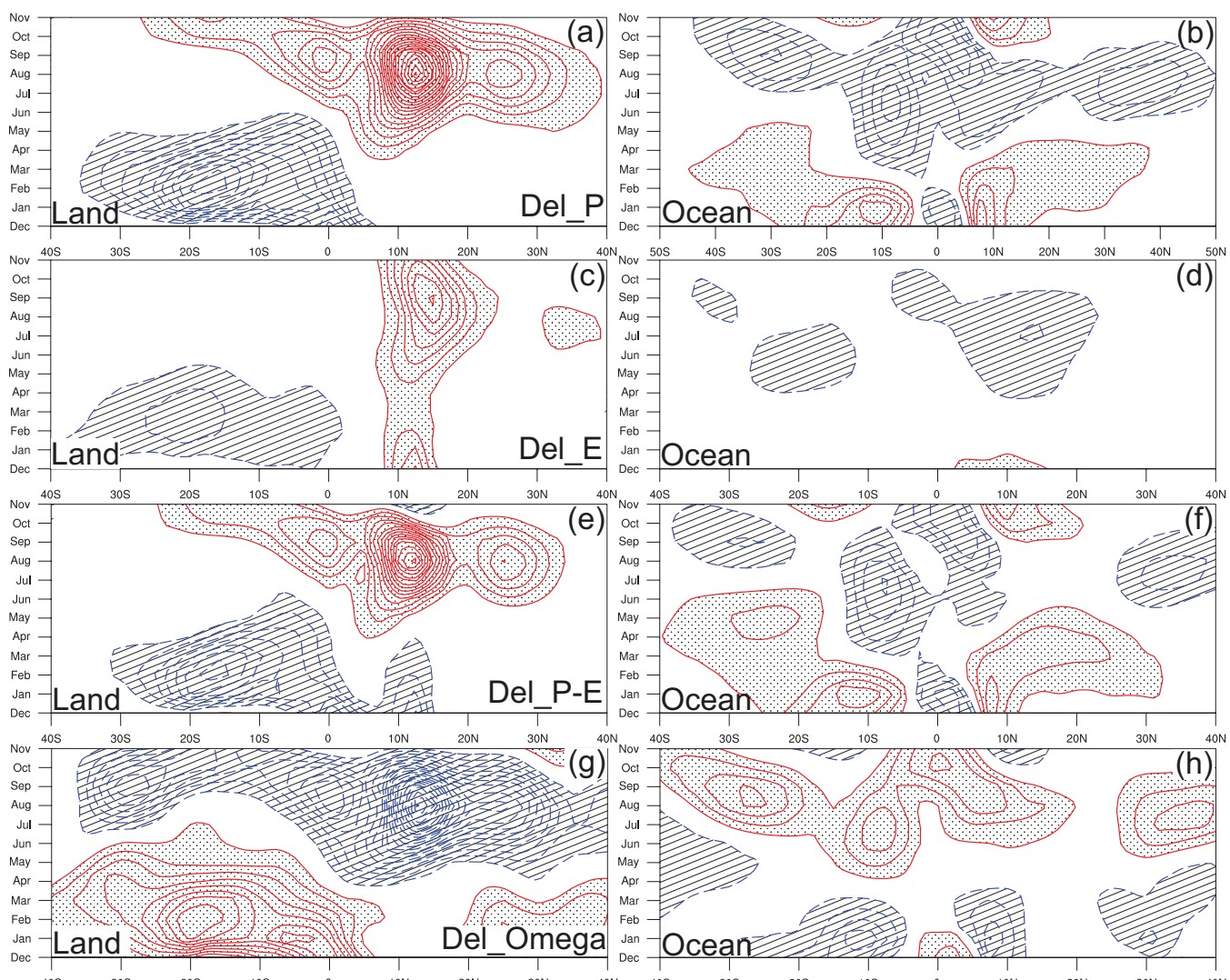

**Figure 5.** (a) and (b) are PMIP4-CMIP6 multimodel seasonal evolution of precipitation changes over land and ocean, respectively. (c) and (d) are as (a) and (b), but for the evaporation changes over land and ocean, respectively. (e) and (f) are as (a) and (b), but for the $P - E$ changes over land and ocean, respectively. The contour interval in (a)-(f) is 0.1 mm/day, with red (blue) contours representing positive (negative) values. (g) and (h) are as (a) and (b), but for the changes of $\omega$ (500 hPa) over land and ocean, respectively, with an interval $0.2 \cdot 10^{-2} \ Pa \ s^{-1}$.

that terrestrial hydrological cycle changes are predominantly driven by summer season dynamics with an amplified inter-hemispheric contrast during the mid-Holocene.

Compared to terrestrial hydrological cycle changes, the seasonal evolution of hydrological changes over ocean shows no pronounced inter-hemispheric asymmetry during the mid-Holocene (Figures 5b, 5d, 5f, and 5h). In the marine tropics, both

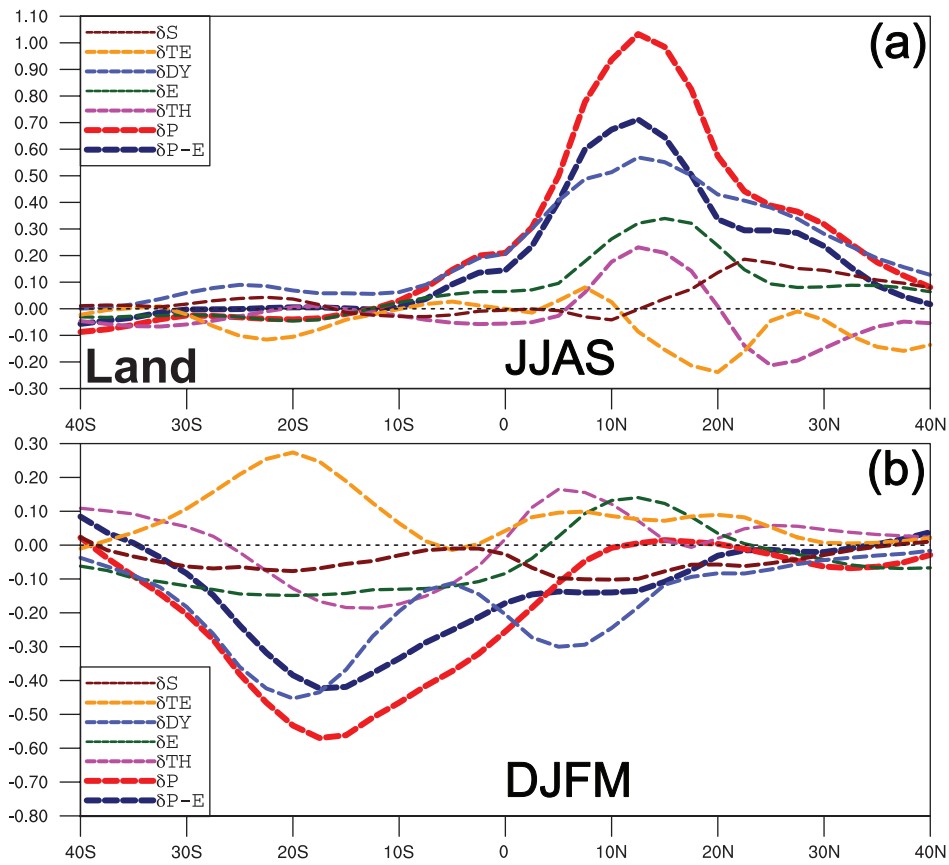

**Figure 6.** PMIP4-CMIP6 multimodel seasonal and zonal mean changes of moisture budget over land in (a) June-to-September (JJAS) and (b) December-to-March (DJFM). The unit is mm/day.

precipitation and runoff decline throughout the annual cycle (Figures 5b, 5f). Meanwhile, in both the northern and southern
subtropical oceans, precipitation and runoff increase during DJFM but decrease in JJAS (Figures 5b, 5f), resulting in a small positive annual-mean anomaly driven by enhanced water vapor convergence (Figure 4c; Bian and Räisänen (2024)). Another notable land–ocean contrast lies in evaporation: while terrestrial evaporation exhibits marked inter-hemispheric differences, oceanic evaporation declines across the annual cycle in both hemispheres (Figures 5c, 5d).

    Further analysis of the seasonal evolution of moisture budget between land and ocean indicates that the dynamic term ($\delta DY$)
is the primary driver of terrestrial precipitation changes in the Northern Hemisphere during JJAS months (Figure 6a), particularly in the Northern Hemisphere monsoonal regions as discussed in D'Agostino et al. (2019). Strengthened mean atmospheric flow (i.e. $\delta DY$) increases moisture convergence, thereby reinforcing the Northern Hemisphere monsoons and monsoonal rainfall during the mid-Holocene (D'Agostino et al., 2019; D'Agostino et al., 2020; Bian and Räisänen, 2024). Conversely, the dynamic term ($\delta DY$) is also the dominant factor in the reduction of terrestrial precipitation in the Southern Hemisphere during
DJFM months (Figure 6b). Unlike the annual moisture budget variations depicted in Figure 4c, the transient eddy flux term

($\delta TE$) plays a compensatory role in the Northern Hemisphere summer terrestrial precipitation changes. Moreover, both hemispheres exhibit relatively minor changes in terrestrial moisture budgets during winter seasons (Figures 6a, 6b), indicating that shifts in the terrestrial hydrological cycle are primarily driven by summer dynamics during the mid-Holocene.

Figures 4c, 4d, and 4f further illustrate that the increased terrestrial rainfall is closely linked to positive anomalies in atmospheric energy divergence ($\nabla \cdot \overline{\mathbf{F}_a} > 0$) over the Northern Hemisphere. This is primarily driven by enhanced vertical moist static energy (MSE) advection over land (Terms III and IV), while changes in horizontal MSE advection (Terms I and II) and eddy MSE flux (Term V) have a compensating influence, as illustrated in Figure 4f.

For the $\delta \nabla \cdot \overline{\mathbf{F}_a} > 0$ and associated changes in MSE vertical advection over land, two physical pathways are relevant. First, changes in radiative forcing result in increased net radiation ($\delta \overline{R_a} > 0$) in the Northern Hemisphere during spring and summer (Brierley et al., 2020). This alters the net energy flux divergence ($\delta \nabla \cdot \overline{\mathbf{F}_a}$), which further influences the temperature and moisture distributions and affects the MSE gradient and advection over land during the mid-Holocene (Figures 4b and 4d). However, in the annual mean, orbital forcing is symmetric between the two hemispheres, while the change in the annual mean atmospheric radiation balance is asymmetric, with a slight positive anomaly ($\delta \overline{R_a} > 0$) from $10°N$ to $30°N$ for land and ocean (Figure 4b), and $15°N$ to $40°N$ for land alone (Figure 4d). This suggests that factors other than the direct orbital forcing are also important.

We further analyze the separate contributions of the shortwave (SW) and longwave (LW) flux components to the atmospheric radiation balance (Figure A1). In the Northern Hemisphere, enhanced atmospheric absorption of SW radiation ($\delta R_s\_SW$), which is primarily due to increased atmospheric water vapor and cloudiness, contrasts with reduced SW absorption in the Southern Hemisphere (Figures A1a, A1c). Additionally, changes in cloudiness and water vapor could affect the rate of longwave cooling and then lead to changes in radiative transfer and heating process. During the mid-Holocene, reduced atmospheric LW cooling (i.e. $-\delta R_a\_LW$; Figures A1b, A1d) further increases the atmospheric radiation balance ($\delta R_a$) in the Northern Hemisphere, though this effect plays a secondary role in creating the hemispheric asymmetry. Therefore, the combination of reduced surface SW radiation ($\delta R_s\_SW$; Figures A1a and A1c) and increased $\delta R_a\_SW$ (i.e. $\delta RTOA\_SW - \delta R_s\_SW$; Figures A1a, A1c) in the Northern Hemisphere suggests that increased cloudiness and water vapor play an important role.

Furthermore, changes in the annual net surface energy budget ($\delta(R_s - SH_0 - LE)$) over land are minimal due to the small heat capacity of the land surface, but over the ocean, shifts in heat transport disturb this balance (Figure A2). Specifically, there are negative anomalies of $\delta(R_s - SH_0 - LE)$ in the Northern Hemisphere tropics and smaller positive anomalies in the Southern Hemisphere (Figure A2). This suggests that ocean currents are transferring more heat from the Southern to the Northern Hemisphere, thereby amplifying the inter-hemispheric asymmetry in the ocean-to-atmosphere energy transfer and annual atmospheric radiation balance during the mid-Holocene. Delineating why the negative $\delta(R_s - SH_0 - LE)$ anomalies in the Northern Hemisphere tropics are larger than the positive anomalies in the Southern Hemisphere tropics would require further investigation.

Second, strong positive latent heat flux anomalies over land significantly contribute to enhanced vertical MSE advection (Figures 4d, 4f). As moist air from the lower levels rises and cools, the latent heat released by the condensation increases the buoyancy of the air parcel, promoting atmospheric instability and stronger updrafts, which facilitates the development

of convective systems (Hill et al., 2017; Adames and Maloney, 2021). Additionally, Figure 4c shows enhanced moisture convergence (P-E) anomalies, combined with increased vertical MSE advection (Figure 4f), resulting in enhanced terrestrial moist convection and precipitation in the Northern Hemisphere during this period. In contrast, reduced rising motion in the Southern Hemisphere causes negative anomalies in vertical MSE advection, suppressing moist convection and precipitation over land during the mid-Holocene.

Figure 4f further shows that enhanced vertical MSE advection over land is primarily due to stronger rising motion (Term IV) within the northern Hadley cell region (Figure 2b). Changes in the vertical MSE gradient profile (Term III in Eq. (9)), play a secondary role in promoting vertical MSE advection (Figure 4g). Furthermore, the spatial distributions of Terms III and IV further indicate that positive anomalies in vertical MSE advection are predominantly concentrated in monsoonal regions of the Northern Hemisphere, playing a key role in promoting terrestrial moist convection and increasing monsoonal rainfall (D'Agostino et al., 2019; Bian and Räisänen, 2024), as illustrated in Figures 4g and 4h.

In summary, the terrestrial hydrological cycle changes are primarily due to summer season dynamics with an amplified inter-hemispheric contrast during the mid-Holocene, while both hemispheres exhibit minor changes in winter seasons. In contrast to its modest contribution to the annual moisture budget changes (Figure 4c), the dynamic term ($\delta DY$) is the primary driver of summer terrestrial precipitation changes in both hemispheres, while the transient eddy flux term ($\delta TE$) has a compensating effect. Furthermore, positive anomalies in vertical MSE advection over land due to stronger upward motion create favorable conditions for moist convection and precipitation (Back and Bretherton, 2006; Chou et al., 2013; Hill et al., 2017; Hill, 2019; Geen et al., 2020; Bombardi and Boos, 2021), particularly evident in the Northern Hemisphere tropics and adjacent subtropical regions during the mid-Holocene.

## 3.4 Impact of hydrological changes on land aridity

The northward ITCZ migration and associated Hadley cell changes can further lead to shifts in regional climates, altering terrestrial dryness across the tropics and subtropics and resulting in changes to regional drylands during the mid-Holocene (Diaz and Bradley, 2004; Lu et al., 2007; Lau and Kim, 2015; Liu et al., 2019; Lionello et al., 2024). We further assess the influence of terrestrial hydrological changes on land aridity during the mid-Holocene by analyzing the climatological moisture availability index (alpha), aridity index (AI), and runoff ratio.

The alpha index is defined as the ratio of annual evaporation to annual potential evaporation (PET), and the runoff ratio is defined as the proportion of annual runoff to annual precipitation. The commonly used AI is defined as the ratio of annual PET to annual precipitation, typically calculated using the Penman-Monteith (PM_AI) reference model for PET (Allen et al., 1998; Feng and Fu, 2013). However, recent studies have shown limitations with the PM_AI method for calculating PET, noting that it can significantly overestimate projected aridity changes (Greve et al., 2019; Yang and Roderick, 2019). Milly and Dunne (2016) found that the PM_AI based method often overpredicts changes in evaporation over well-watered surfaces in climate models. In this study, we employ a net surface radiation ($R_s$) based method for calculating PET, using estimates of $R_s$ as the energy constraint on evaporation (Milly and Dunne, 2016; Yang and Roderick, 2019; Yang et al., 2019; Greve et al., 2019).

Although this is a simplified approach to estimate PET, there is a reasonable empirical correspondence between PET and net surface radiation (Milly and Dunne, 2016; Greve et al., 2019), as $PET = 0.8 \frac{R_s}{L}$.

To quantify the influence of precipitation, evaporation, and PET on land aridity changes, we decompose the AI and runoff ratio index changes by Taylor expansion:

$$\delta(\text{AI}) = \delta\left(\frac{\overline{PET}}{\overline{P}}\right) = \frac{1}{\overline{P}}\delta\overline{PET} - \frac{\overline{PET}}{\overline{P}^2}\delta\overline{P} + Ro_1 \tag{13}$$

and

$$\delta(\text{runoff ratio}) = \delta\left(\frac{\overline{P} - \overline{E}}{\overline{P}}\right) = -\frac{1}{\overline{P}}\delta\overline{E} + \frac{\overline{E}}{\overline{P}^2}\delta\overline{P} + Ro_2 \tag{14}$$

where $\delta(\cdot)$ denotes $(\cdot)_{MH} - (\cdot)_{PI}$.

$\frac{1}{\overline{P}}\delta\overline{PET}$ is the contribution of PET changes to AI changes, and $-\frac{\overline{PET}}{\overline{P}^2}\delta\overline{P}$ is the contribution of precipitation changes to AI changes. $-\frac{1}{\overline{P}}\delta\overline{E}$ is the contribution of evaporation changes to runoff ratio changes, and $\frac{\overline{E}}{\overline{P}^2}\delta\overline{P}$ is the contribution of precipitation changes to runoff ratio changes. $Ro_1$ and $Ro_2$ are residual terms representing non-linear effects (not discussed in this study).

The reconstructed precipitation changes are closely linked to changes in the reconstructed moisture availability index (alpha) based on pollen records from the mid-Holocene (Bartlein et al., 2011; Herbert and Harrison, 2016; Lowry and McGowan, 2024), as shown in Figures 1f and 7a. Furthermore, the comparison of the spatial distribution of changes in the simulated alpha index (Figure 7a), AI (Figure 7c), and runoff ratio (Figure 7d) generally indicates that monsoonal regions experience wetter conditions with reduced terrestrial aridity in the Northern Hemisphere, leading to a contraction of regional drylands during the mid-Holocene (Figures 8a-8d). Additionally, despite simulated circulation changes that generally favor increased moist convection and rainfall in the Northern Hemisphere, pollen records in eastern North America and central Asia indicate a drier climate than at present (Bartlein et al., 2011). Analysis of the RHS terms in Eqs. (13) and (14) suggests that increased precipitation is the primary factor driving wetter conditions and the reduction of terrestrial aridity in the Northern Hemisphere (Figures 7g, 7h), while changes in PET and evaporation have a compensatory effect (Figures 7e, 7f). Specifically, changes in evaporation contribute to drier conditions in eastern North America and central Asia, as reflected in the alpha index changes shown by pollen records in Figure 7a, while changes in AI and runoff ratio fail to capture these regional transitions to drier climate.

In the Southern Hemisphere, both simulations and reconstructions indicate a generally drier climate during the mid-Holocene (Bartlein et al., 2011; Prado et al., 2013; Herbert and Harrison, 2016; Lim et al., 2016; Liu et al., 2019; Lowry and McGowan, 2024), resulting in increased terrestrial aridity and the expansion of regional drylands in South America, southeastern Africa, and northern Australia (Figures 7c-7d, 8e-8g). However, simulated land dryness changes in Australia, represented by the alpha index (Figure 7b), AI (Figure 7c), and runoff ratio (Figure 7d), still show spatial discrepancies when compared to the reconstructions in Figure 7a, as noted earlier by Liu et al. (2019) and Lowry and McGowan (2024). Although the PMIP4 simulation results in Figure 1b indicate drier conditions in northern Australia, only a few pollen records from near the Cape York Peninsula and northeastern Australia support this trend while others do not (Lowry and McGowan, 2024). Additional

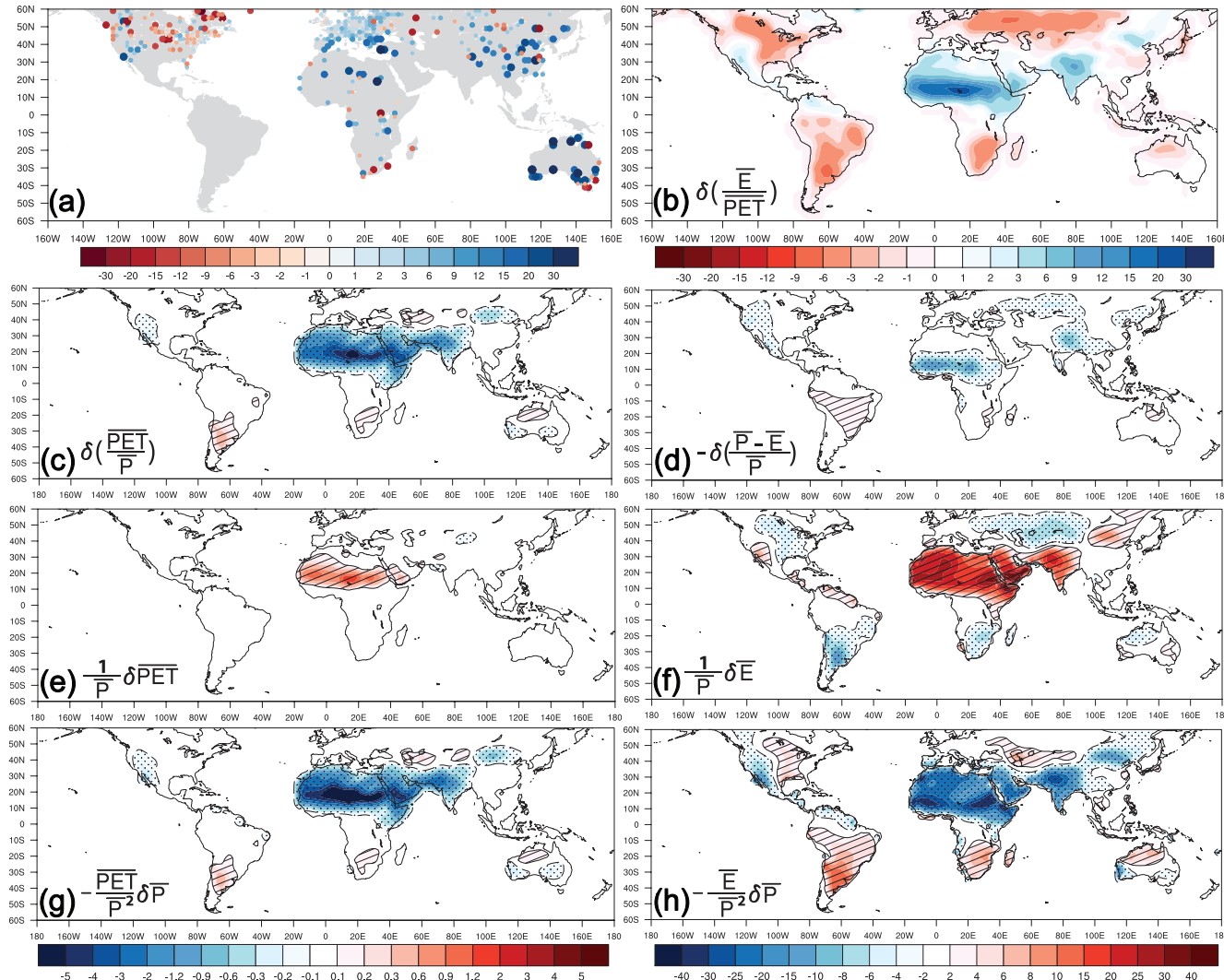

**Figure 7.** (a) Reconstructed moisture availability index (alpha) changes (%) from pollen proxies (Bartlein et al., 2011; Herbert and Harrison, 2016; Lowry and McGowan, 2024). Sites with blue color indicate wetter soil conditions during the mid-Holocene compared to present-day (0 ka) proxy records, and red ones for drier soil conditions. (b) Changes of alpha from PMIP4 simulations. (c) Changes of annual AI from PMIP4 simulations. (d) as (c), but for the reversed changes (%) of annual runoff ratio from the PI to the MH period. (e), (g) are the first and second RHS terms of Eq. (13) for AI changes. (f) and (h) are the contributions of the first and second RHS terms in Eq. (14) to the runoff ratio changes (%), with signs reversed.

reconstructions from northwestern Australia reveal reduced fluvial activity and dune reactivation (Reeves et al., 2013), and a reduction in rain-forest taxa and mangrove (Proske et al., 2014; Field et al., 2017), indicating a contraction of the local ITCZ that contributes to reduced Australian monsoon activity and precipitation (Reeves et al., 2013; Brierley et al., 2020; D'Agostino et al., 2020). Furthermore, part of pollen records from southeastern Australia (Figure 7a) indicate wetter conditions during the

mid-Holocene (Herbert and Harrison, 2016; Lowry and McGowan, 2024), which are likely caused by precipitation-driven reduced terrestrial aridity shown in AI and runoff ratio changes (Figures 7g, 7h, 8g). Complementary paleo-hydrologic data from the Oz-INTIMATE series further indicate wetter conditions in temperate Australia, with enhanced river discharge and increased precipitation during the mid-Holocene (Petherick et al., 2013; Lowry and McGowan, 2024). However, as discussed in Section 3.1, the PMIP4 simulations do not consistently capture these significant precipitation changes in reconstructions (Figure 1f).

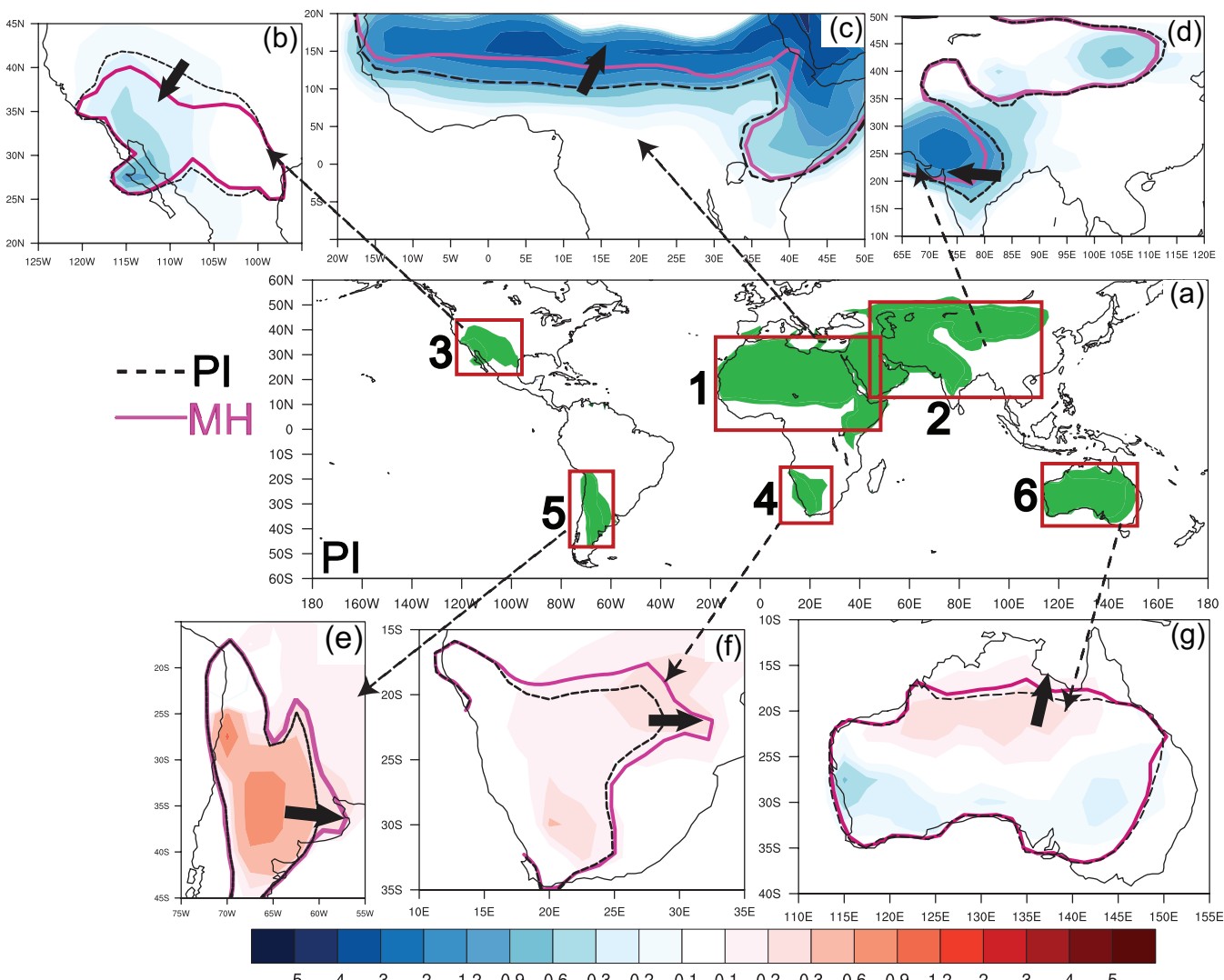

**Figure 8.** (a) Annual mean dryland distribution by aridity index (AI) in the PI period. The shaded areas denote drylands (semi-desert and desert) with AI values equal and larger than 1.6, adapted from Nicholson (2011). (c)-(g) are dryland shifts between the PI and MH periods. The colored values in (c)-(g) denote AI anomalies.

As the alpha index can be influenced by variations in temperature and precipitation, its changes in simulations and reconstructions are generally less consistent and pronounced than those in precipitation. The temperature-related changes in evaporation may cause significant changes in the alpha index even in areas with minimal changes in precipitation. Therefore, potential uncertainties in the pollen-based reconstructed alpha index should also be considered (Bartlein et al., 2011). Other potential biases in quantitative climate reconstructions based on pollen data include the impact of local vegetation on the pollen assemblages, differences in pollen productivity and dispersal, and the lack of suitable sedimentary environments, notably lakes, in the arid regions in tropics and subtropics (Birks et al., 2011; Huntley, 2012). An additional source of uncertainty is the extent to which changes in the AI effectively impact soil moisture and land aridity (Greve et al., 2019; Scheff et al., 2021, 2022). This highlights the importance of comparing changes in the alpha index, AI, and runoff ratio to assess land aridity changes during the mid-Holocene. Recognizing inherent uncertainties in both climate simulations and reconstructions is essential.

## 4   Summary and discussion

This study investigates the impacts of northward ITCZ migration on multiple Hadley cell characteristics and their combined influence on the global hydrological cycle and terrestrial aridity during the mid-Holocene, approximately 6,000 years ago. Known as the "Green Sahara", this era was marked by significant climatic changes, with arid regions like the Sahara experiencing wetter conditions. We apply different precipitation metrics to quantify the global ITCZ position, and the widths and strengths of the Hadley cell are based on two mass streamfunction metrics, using nine simulations from the PMIP4-CMIP6 archive.

Our findings indicate that during the mid-Holocene, increased energy input to the Northern Hemisphere atmosphere, ultimately driven by orbital forcing but modulated by processes internal to the climate system, shifted the global and annual ITCZ northward by 0.2° and 0.3° using two different metrics. Those shifts align with mid-Holocene proxy evidence indicating a northward ITCZ migration up to 1° (Haug et al., 2001; McGee et al., 2014). The northward ITCZ migration is accompanied with multiple changes in the Hadley cell that differ from those changes under the current greenhouse gas (GHG)-induced warming climate (Lu et al., 2007; Lau and Kim, 2015; Byrne et al., 2018; Grise and Davis, 2020; Pikovnik et al., 2022; Lionello et al., 2024; Wu et al., 2024). Specifically, the northern Hadley cell width contracted by 1.1° and 0.5° using the two metrics, and its strength reduced by 3.7% and 4.1%, with weaker overturning circulation, stronger rising motion, and a wetter troposphere during the mid-Holocene. This contributes to increased precipitation in the tropics and adjacent subtropical regions, particularly in monsoonal regions of the Northern Hemisphere. Conversely, the southern Hadley cell expanded by 1.2° and 0.6° and strengthened by 2.9% and 1.8%, with stronger overturning circulation, reduced rising motion near and to the south of the Equator, and a drier troposphere, contributing to decreased precipitation in the Southern Hemisphere. Under the current GHG-induced warming scenario, the ITCZ narrows and intensifies, leading to the deep-tropics squeeze phenomenon and affecting the Hadley cell by concentrating rising air, strengthening meridional circulation, and causing drier subtropical conditions with expanding drylands (Lu et al., 2007; Lau and Kim, 2015; Lionello et al., 2024). These differences highlight the distinct impacts of orbital forcing and GHG-induced warming on large-scale circulation dynamics and hydrological patterns.

Moisture budget analysis reveals that mid-Holocene precipitation changes are driven by variations in key components: evaporation, moisture advection, and eddy fluxes. Enhanced eddy fluxes significantly contribute to increased terrestrial precipitation across the Northern Hemisphere tropics and adjacent subtropical regions. In contrast, negative Southern Hemispheric precipitation anomalies are primarily attributed to evaporation and dynamic factors. Furthermore, increased radiative heating and terrestrial latent heating in the Northern Hemisphere alter the MSE gradient and advection, intensifying moist convection and precipitation. Specifically, enhanced rising motion leads to positive anomalies in vertical MSE advection over land, further promoting moist convection and precipitation in the Northern Hemisphere. Conversely, weaker rising motion leads to reduced vertical MSE advection in the Southern Hemisphere, suppressing moist convection and precipitation.

The terrestrial hydrological cycle changes are primarily due to summer season dynamics with an amplified inter-hemispheric contrast during the mid-Holocene, while both hemispheres exhibit minor changes in winter seasons. In contrast to its modest contribution to the annual moisture budget changes, the dynamic term ($\delta DY$) is the primary driver of summer terrestrial precipitation changes in both hemispheres, while the transient eddy flux term ($\delta TE$) has a compensating effect. For the seasonal evolution of hydrological changes over ocean, there is no pronounced inter-hemispheric asymmetry during the mid-Holocene. In the marine tropics, both precipitation and runoff decline throughout the annual cycle, while precipitation and runoff increase during DJFM months but decrease in JJAS months in subtropical oceans of both hemispheres.

Although orbital forcing during the mid-Holocene was symmetric around the equator in the annual mean, it indirectly drove hemispherically asymmetric changes in annual atmospheric radiation balance. In the Northern Hemisphere, reduced surface shortwave radiation ($\delta R_s\_SW$) alongside increased atmospheric shortwave absorption $\delta R_a\_SW$ indicates that enhanced cloudiness and water vapor play key roles. Additionally, the net surface energy budget ($\delta(R_s - SH_0 - LE)$) increases over the Southern Hemisphere oceans but decreases over the Northern Hemisphere oceans. This suggests increased ocean heat transport from the Southern to the Northern Hemisphere, which likely further increased ocean-to-atmosphere heat transfer and atmospheric radiation balance in the Northern relative to the Southern Hemisphere.

The northward ITCZ migration and the associated changes in Hadley cell significantly influence regional climates by altering terrestrial dryness across the tropics and subtropics. Both simulations and reconstructions generally agree on the wetter conditions in the Northern Hemisphere (Bartlein et al., 2011), supporting the conclusion of a northward-shifted ITCZ and a narrowed, weakened northern Hadley Cell. These changes contribute to reduced land aridity in tropical and adjacent subtropical regions of the Northern Hemisphere. Additionally, pollen records and other types of proxies generally indicate a drier climate in the Southern Hemisphere, consistent with the simulated expansion of the southern Hadley cell and reduced moisture convergence. However, notable discrepancies remain between simulations and reconstructions in some regions, particularly in Australia (Krause et al., 2019; Liu et al., 2019; D'Agostino et al., 2020; Lowry and McGowan, 2024).

The core of energetic theory assumes that the atmospheric cross-equatorial energy flux transport is primarily driven by the zonal mean meridional circulation (i.e. Hadley cell), with contributions from stationary and transient eddy fluxes and changes in gross moist stability (GMS) considered negligible (Adam et al., 2016; Donohoe and Voigt, 2017; Wei and Bordoni, 2018; Hill, 2019; Kang, 2020). However, both observations and climate simulations reveal that eddy fluxes play a non-negligible role in energy and moisture budgets in the tropical atmosphere (Roberts et al., 2017; Byrne et al., 2018; Geen et al., 2020; Kang,

2020). For the mid-Holocene climate, our results indicate that stationary and transient eddies changes significantly influence on cross-equatorial energy flux transport. This means energetic constraint theories linking the ITCZ and Hadley cell have limitations in explaining ITCZ changes and their nonlinear dependency on the Hadley cell. Roberts et al. (2017) highlights that there is insufficient evidence to support the Hadley cell as the primary driver of tropical rainfall and energy transport, particularly when considering the effect of stationary and transient energy fluxes. Moreover, energetic theory for the zonal-

mean framework cannot fully account for the relationship between regional rainfall changes and meridional migrations of the ITCZ, as the direction and magnitude of regional rainfall band migration vary significantly by longitude (Roberts et al., 2017; Atwood et al., 2020). Additionally, even when Hadley cell mean meridional circulation dominates cross-equatorial atmospheric energy transport, the energetic framework may still be insufficient, as changes in GMS can also make a contribution (Wei and Bordoni, 2018; Kang, 2020).

Although all nine simulations applied in this study show a northward migration and zonal precipitation belt during the mid-Holocene, intermodel differences in precipitation changes may introduce uncertainties when these results are compared with paleoclimate reconstructions (McGee et al., 2014; Byrne et al., 2018; Bian and Räisänen, 2024). Moreover, the PMIP4 simulations do not account for the dynamic feedbacks caused by the climate-vegetation coupling and emissions of multiple types of aerosols, leaving uncertainty about how a fully coupled atmosphere-ocean-vegetation-aerosol framework might influence

the simulated global hydrological cycle and terrestrial aridity during the mid-Holocene (Pausata et al., 2016; Tierney et al., 2017; Hopcroft and Valdes, 2019; Brierley et al., 2020; Pausata et al., 2020; Bian et al., 2023; Jungandreas et al., 2023). In the "Green Sahara" period, multiple reconstructions have shown significantly increased precipitation, amplified vegetation cover, and reduced dust emissions (Claussen et al., 2017; Tierney et al., 2017; Pausata et al., 2020; Brierley et al., 2020; Thompson et al., 2022; Kaufman and Broadman, 2023). These findings highlight the critical role of vegetation and aerosols feedbacks

in shaping mid-Holocene climate. Therefore, exploring how global-scale changes in dynamic vegetation cover and aerosol emissions interact to modulate ITCZ migration, Hadley Cell adjustments, and the evolution of the global hydrological cycle is crucial for future mid-Holocene study.

Overall, this study underscores the complex interactions among orbital forcing, the ITCZ, the Hadley cell, global hydrological cycle, and terrestrial aridity during the mid-Holocene. While climate models provide valuable insights into these dynamics,

it is crucial to acknowledge their limitations and the uncertainties associated with both simulations and reconstructions. Continued research and model development are needed to improve our understanding of past climate changes and their implications for the future.

# Appendix A

**Table A1.** List of the nine PMIP4-CMIP6 models.

|   | Model | Realization | Used years | Model reference |
|---|-------|-------------|------------|-----------------|
| 1 | EC-Earth3-LR | r1i1p1 | 100 | (Wyser et al., 2020) |
| 2 | CESM2 | r1i1p1 | 100 | (Gettelman et al., 2019) |
| 3 | IPSL-CM6A-LR | r1i1p1 | 100 | (Boucher et al., 2020) |
| 4 | ACCESS-ESM1-5 | r1i1p1 | 100 | (Ziehn et al., 2020) |
| 5 | AWI-ESM-1-1-LR | r1i1p1 | 100 | (Sidorenko et al., 2015) |
| 6 | GISS-E2-1-G | r1i1p1 | 100 | (Kelley et al., 2020) |
| 7 | MPI-ESM1-2-LR | r1i1p1 | 100 | (Mauritsen et al., 2019) |
| 8 | MRI-ESM2-0 | r1i1p1 | 100 | (Yukimoto et al., 2019) |
| 9 | NorESM2-LM | r1i1p1 | 100 | (Seland et al., 2020) |

**Table A2.** Digital Object Identifier (doi) of the nine PMIP4-CMIP6 models.

|   | Model | midHolocene | piControl |
|---|-------|-------------|-----------|
| 1 | EC-Earth3-LR | https://dx.doi.org/10.22033/ESGF/CMIP6.4801 | https://dx.doi.org/10.22033/ESGF/CMIP6.4847 |
| 2 | CESM2 | https://dx.doi.org/10.22033/ESGF/CMIP6.7674 | https://dx.doi.org/10.22033/ESGF/CMIP6.7733 |
| 3 | IPSL-CM6A-LR | https://dx.doi.org/10.22033/ESGF/CMIP6.5229 | https://dx.doi.org/10.22033/ESGF/CMIP6.5251 |
| 4 | ACCESS-ESM1-5 | https://dx.doi.org/10.22033/ESGF/CMIP6.13704 | N/A |
| 5 | AWI-ESM-1-1-LR | https://dx.doi.org/10.22033/ESGF/CMIP6.9332 | https://dx.doi.org/10.22033/ESGF/CMIP6.9335 |
| 6 | GISS-E2-1-G | https://dx.doi.org/10.22033/ESGF/CMIP6.7225 | https://dx.doi.org/10.22033/ESGF/CMIP6.7380 |
| 7 | MPI-ESM1-2-LR | https://dx.doi.org/10.22033/ESGF/CMIP6.6644 | https://dx.doi.org/10.22033/ESGF/CMIP6.6675 |
| 8 | MRI-ESM2-0 | https://dx.doi.org/10.22033/ESGF/CMIP6.6860 | https://dx.doi.org/10.22033/ESGF/CMIP6.6900 |
| 9 | NorESM2-LM | https://dx.doi.org/10.22033/ESGF/CMIP6.8079 | https://dx.doi.org/10.22033/ESGF/CMIP6.8217 |

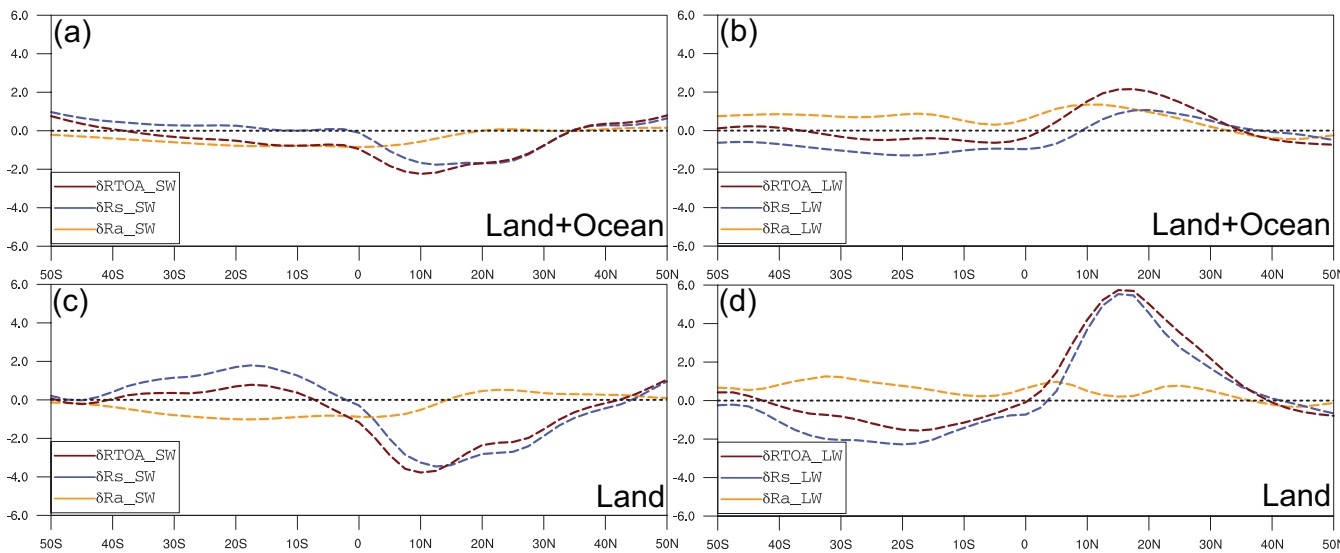

**Figure A1.** Individual contributions of the shortwave (a) and longwave (b) flux components to the changes of annual atmospheric radiation balance from the PI to the MH period. (c), (d) are as (a), (b) respectively, but for land only. The unit is $Wm^{-2}$. All results are based on PMIP4-CMIP6 multimodel averages.

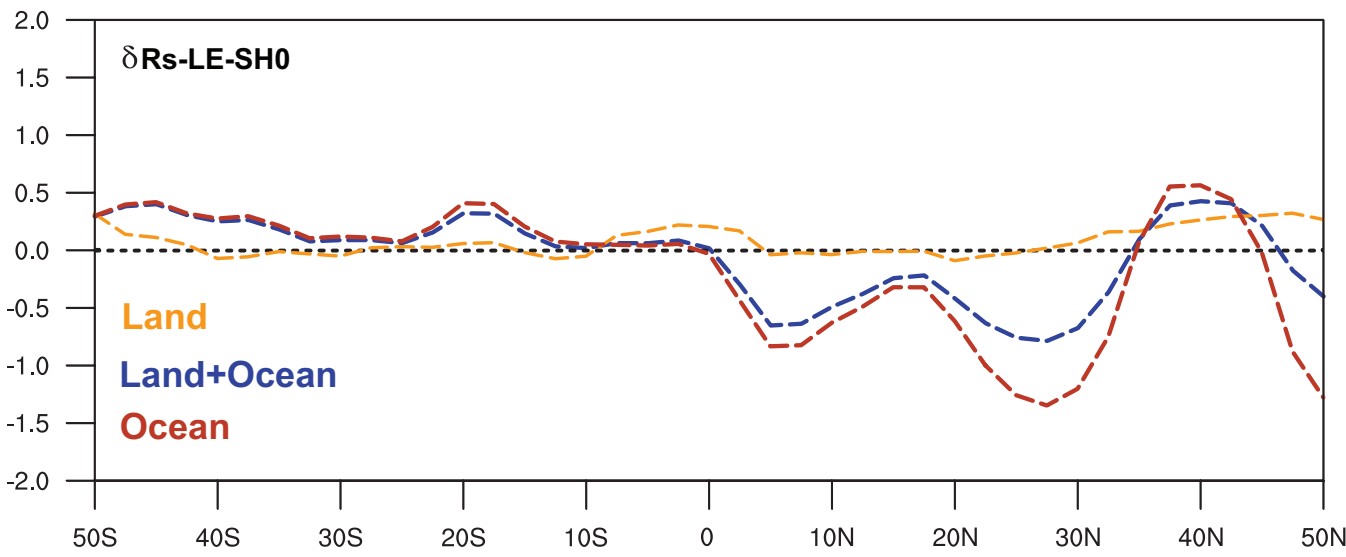

**Figure A2.** Changes in the annual net surface energy budget ($\delta(R_s - SH_0 - LE)$) from the PI to the MH period (unit: $Wm^{-2}$). All results are based on PMIP4-CMIP6 multimodel averages.

## Appendix B: Energy balance and ITCZ latitude

The energetic constraint theory suggests that changes of the ITCZ latitude ($\phi_{ITCZ}$) are proportional to the change of annual cross-equatorial energy transport ($\overline{F}$) (Kang et al., 2008; Adam et al., 2016; Donohoe and Voigt, 2017; Roberts et al., 2017; Wei and Bordoni, 2018; Kang, 2020; Geen et al., 2020)

$$\delta\phi_{ITCZ} \propto -\delta\overline{F} \tag{B1}$$

where the negative anomaly of cross-equatorial energy transport (i.e. $\delta\overline{F} < 0$) implies a positive shift in $\delta\phi_{ITCZ}$ (i.e. a north-
565 ward displacement).

Furthermore, the annual mean atmospheric meridional energy transport across latitude $\phi$ as

$$\overline{F}(\phi) = \frac{2\pi a \cos\phi}{g} \int_0^{p_s} \underbrace{[\overline{v}][\overline{MSE}]}_{MMC} + \underbrace{[\overline{v}^*\overline{MSE}^*]}_{SE} + \underbrace{[\overline{v'MSE'}]}_{TE} dp \tag{B2}$$

where $MMC$, $SE$, and $TE$ refer to the contributions of the mean meridional circulation, stationary eddies, and transient eddies (Adam et al., 2016; Roberts et al., 2017; Kang, 2020) .

Under long-term time means, energy balance holds in the atmosphere. Therefore, the meridional energy flux at latitude $\phi$ equals to the net effect of the source terms, integrated from the South Pole and to the latitude considered:

$$\overline{F}(\phi) = 2\pi a \int_{-\frac{\pi}{2}}^{\phi} ([\overline{R_a}] + [\overline{SH_0}] + L[\overline{E}]) \cos\phi d\phi \tag{B3}$$

The MMC and SE contributions to the atmospheric energy transport are calculated from Eq. B2 using monthly mean data. The TE contribution is obtained as a residual using the total transport calculated from Eq. B3: $TE = \overline{F} - MMC - SE$.

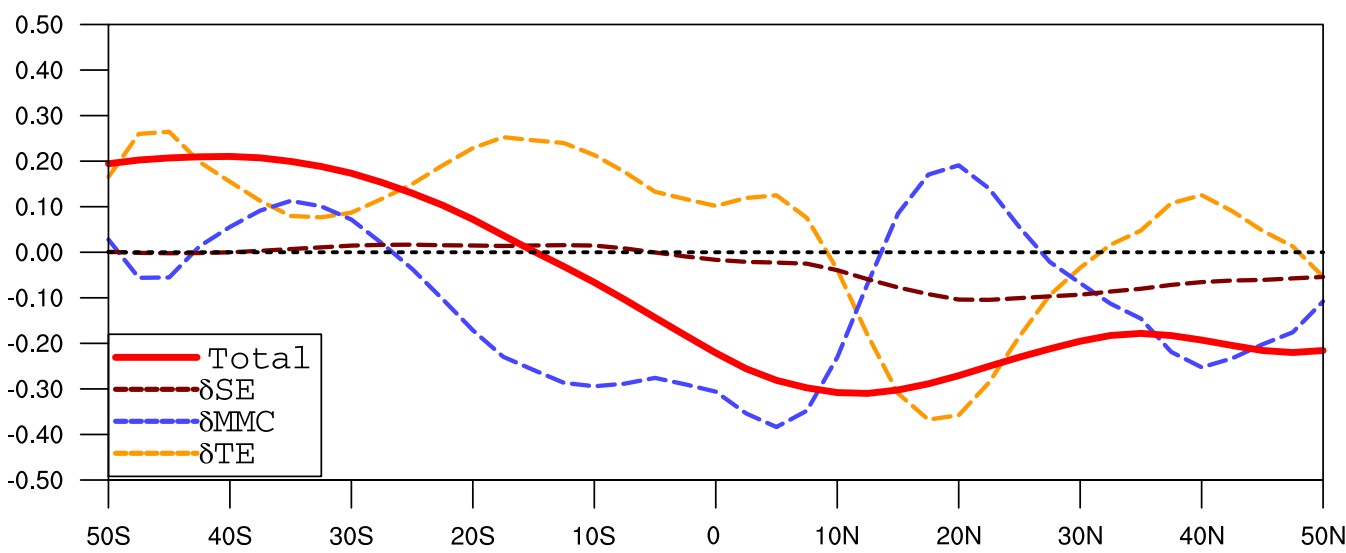

**Figure B1.** Changes in the annual atmospheric meridional energy transport by the mean meridional circulation (MMC), stationary eddies (SE), and transient eddies (TE) from the PI to the MH period (unit: $PW$). See Eqs. B2-B3 in Appendix B for further details. All results are based on PMIP4-CMIP6 multimodel averages.

*Data availability.* The PMIP4-CMIP6 simulations are available at https://esgf-node.llnl.gov/. The GPCP monthly precipitation dataset is available at https://downloads.psl.noaa.gov/Datasets/gpcp.

For alpha index and MAP pollen proxy databases in Bartlein et al. (2011), the available links are https://www.ncei.noaa.gov/pub/data/paleo/pollen/recons/bartlein2010/alpha_delta_06ka_ALL_grid_2x2.nc and https://www.ncei.noaa.gov/pub/data/paleo/pollen/recons/bartlein2010/map_delta_06ka_ALL_grid_2x2.nc. Pollen dataset for alpha index and MAP in Australia (Herbert and Harrison, 2016; Lowry and McGowan, 2024) are detailed in Tables S2 and S3 of the Supplementary material. Dataset for multiple types of proxy in South America (Prado et al., 2013) is available at https://doi.org/10.1594/PANGAEA.820035

*Author contributions.* J.B. conceived the study and wrote the manuscript. All authors participated in the manuscript's discussion and revision.

*Competing interests.* The authors declare no competing interests

*Acknowledgements.* J.B. is supported by the Flagship grant (337549) from the Academy of Finland. Computing resources for this research were provided by CSC - IT Center for Science.

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
