# Peer review of "Mid-Holocene ITCZ migration: connection with Hadley cell dynamics and impacts on terrestrial hydroclimate"

_EGUsphere, 2024_

## Author Comment (AC1)

**Review of "Mid-Holocene ITCZ migration: impacts on Hadley cell dynamics and terrestrial hydroclimate" by Jianpu Bian, Jouni Räisänen, and Heikki Seppä.**

This paper presents an analysis of the PMIP4 mid-Holocene (MH) simulations using a set of metrics to quantify changes in the ITCZ and Hadley Cell edge and extent. The paper also includes a broader discussion of the aridity and atmospheric circulation changes in the MH simulations and comparison with available proxy records. The paper is generally clearly written and describes the experiments and results thoroughly. There is a detailed comparison of different methods to define the Hadley Cell edge and strength and the MSE budget. The results will contribute to understanding of MH climate changes in climate model simulations. Overall the paper is a valuable contribution, presenting new and interesting results. I support publication subject to revisions addressing comments outlined below.

--We sincerely thank the reviewer for the thoughtful and constructive comments, which have significantly improved the quality of our manuscript. We have made substantial revisions and greatly extended the manuscript, with four new figures added for the analysis. Please find our point-to-point responses (in blue) to the reviewer comments (in black).

**General comments:**

My main concern is the focus on the annual mean changes. The key changes in the mid-Holocene are associated with seasonal shifts in insolation due to altered timing of perihelion, so that the seasonal mean anomalies will be larger and easier to interpret. Annual changes may involve offsetting summer and winter changes – for example, see Figure 6 of Brierley et al. (2020) mid-Holocene PMIP paper, where many changes in DJF and JJA precipitation are opposite in sign.

I suggest including analysis of the seasonal changes in addition to annual mean. As there is a clear seasonal change in insolation due to orbital changes, the resulting changes in atmospheric circulation and precipitation will be easier to interpret at seasonal time scales, e.g. for the summer monsoons in each hemisphere. It is also possible that MH minus PI Hadley cell changes will differ in Northern Hemisphere (NH) versus Southern Hemisphere (SH) winter given the strong seasonal orbital forcing anomalies.

--Reply. We agree with this comment and appreciate the suggestion. Although our study primarily focuses on annual changes in global and terrestrial hydroclimate, we acknowledge the importance of seasonal evolution in the hydrological cycle. We have rewritten **Section 3.3** and added three related paragraphs and two new figures (i.e. **Figures 5 and 6** in revised manuscript) to further analyze seasonal hydrological cycle changes and compare them with annual means. Additionally, we have further compared our results with Brierley et al. 2020, D'Agostino et al. 2019, 2020, and other related studies.

- During the mid-Holocene, seasonal anomalies are generally larger than annual changes. **L311-314**: The changes in perihelion precession and altered insolation patterns in the mid-Holocene influenced seasonal and inter-hemispheric thermal contrast between winter and summer, intensifying the seasonal variation in the cross-equatorial Hadley cell and ITCZ (Diaz and Bradley, 2004; Harrison et al., 2014; Claussen et al., 2017; Brierley et al., 2020; Lionello et al., 2024).

- We agree that the annual precipitation changes may involve offsetting summer and winter changes as mentioned in Brierley et al. 2020. In tropics and subtropics in both hemispheres, the global precipitation and P-E anomalies have reversed changes between winter and summer. Regarding the seasonal evolution over land and ocean separately, as shown in **Figure 5**, the marine areas also have contrasting precipitation and P-E anomalies between winter and summer, particularly in subtropics. Over land, annually averaged changes in the terrestrial hydrological cycle primarily reflect the changes in the summer season, with an amplified inter-hemispheric contrast during the mid-Holocene. Land areas in both hemispheres exhibit only minor changes in winter (**Figure 5**).

[revised manuscript text omitted]

■ Our work in preparation for the mid-Holocene asymmetric evolution of Hadley cells shows that the cross-equatorial winter Hadley cell becomes much stronger with increased width during the mid-Holocene, while the summer Hadley cell weakens and shifts northward. The enhanced seasonal contrasts in Hadley cell dynamics further influence hemispheric monsoon patterns as noted in D'Agostino et al., 2019, 2020.

There is also some apparent confusion regarding the annual mean insolation change due to orbital differences at 6ka relative to pre-industrial. For example, in the abstract it is stated that "orbital forcing increased radiative heating in the Northern Hemisphere" and at line 318, the paper mentions "increased solar radiation in the Northern Hemisphere driven by orbital forcing". While this is true for the NH spring/summer, the paper presents annual mean results only.

Obliquity changes at 6ka cause annual mean heating at high latitudes versus tropics in both hemispheres, whereas precession of perihelion causes anomalies in seasonal heating in both NH and SH which cancel out for the annual average. It is therefore confusing to present annual mean results but refer to orbital increases in NH insolation and heating.

There may be an annual mean increase in NH temperature in some models – but there is no figure showing this, and it is not evident in the PMIP4 ensemble mean temperature as shown in Brierley et al. (2020) Figure 1: the NH is mainly cooler except for high northern latitudes. Given this, I cannot see how the results show "increased radiative heating in the Northern Hemisphere" for annual averages. Please clarify this.

--Reply. We appreciate the suggestion and agree that the orbital forcing was discussed in a potentially misleading way in the original manuscript. We have revised those places accordingly in the revision. Furthermore, we have added three paragraphs in **Section 3.3** and two extra figures in the **Appendix** to further clarify the changes in atmospheric radiation balance during the mid-Holocene.

- **L360-367:** We agree that changes in radiative forcing due to orbital parameters result in increased net radiation ($\delta Ra > 0$) in the Northern Hemisphere during spring and summer (Brierley et al., 2020). In the annual mean, orbital forcing is symmetric between the two hemispheres, while the change in the annual mean atmospheric radiation balance is asymmetric, with a slight positive anomaly ($\delta Ra > 0$) from 10°N to 30°N for land and ocean (**Figure 4b**), and 15°N to 40°N for land alone (**Figure 4d**). This suggests that factors other than the direct orbital forcing are also important.

- **L368-377**. We further analyze the separate contributions of the shortwave (SW) and longwave (LW) flux components to the atmospheric radiation balance (**Figure A1**). In the Northern Hemisphere, enhanced atmospheric absorption of SW radiation ($\delta Rs\_SW$), which is primarily due to increased atmospheric water vapor and cloudiness, contrasts with reduced SW absorption in the Southern Hemisphere (**Figures A1a, A1c**). Additionally, changes in cloudiness and water vapor could affect the rate of longwave cooling and then lead to changes in radiative transfer and heating process. During the mid-Holocene, reduced atmospheric LW cooling (i.e. $-\delta Ra\_LW$; **Figures A1b, A1d**) further increases the atmospheric radiation balance ($\delta Ra$) in the Northern Hemisphere, though this effect plays a secondary role in creating the hemispheric asymmetry. Therefore, the combination of reduced surface SW radiation ($\delta Rs\_SW$; **Figures A1a and A1c**) and increased $\delta Ra\_SW$ (i.e. $\delta RTOA\_SW - \delta Rs\_SW$; **Figures A1a, A1c**) in the Northern Hemisphere suggests that increased cloudiness and water vapor play an important role.

- **L378-384**. Furthermore, changes in the annual net surface energy budget ($\delta(Rs - SH0 - LE)$) over land are minimal due to the small heat capacity of the land surface, but over the ocean, shifts in heat transport disturb this balance (**Figure A2**). Specifically, the negative anomalies of $\delta(Rs - SH0 - LE)$ in the Northern Hemisphere tropics seem much larger than the positive anomalies in the Southern Hemisphere tropics (**Figure A2**). This disparity suggests that ocean currents are transferring more heat from the Southern Hemisphere to the Northern Hemisphere, thereby amplifying the inter-hemispheric asymmetry in the ocean-to-atmosphere energy transfer and annual atmospheric radiation balance during the mid-Holocene.

- **L384-385**. Additionally, inherent geographic differences between the hemispheres, particularly in land-sea distribution, may further contribute to this asymmetry, although delineating the precise processes involved remains challenging.

**Specific comments:**

Abstract, line 16: Clarify that the wetter NH and drier SH is mainly over land. (Note that this also may vary when considering seasonal changes – see General Comment above).

--Reply. We agree with this comment and have revised the text accordingly.

Line 59: Use of bilinear interpolation – it is normally better to use conservative regridding for fields such as precipitation and mass streamfunction. Please confirm the choice of regridding does not significantly alter the results.

--Reply. We agree with this comment. We further tested the conservative interpolation method for precipitation and streamfunction metrics and compared them with bilinear methods. The differences among these results are generally small and non-significant. We have revised and confirmed this issue in Section 2.1. Please find further details in **L103-105**.

Line 130: As discussed in General Comment above, there would be some benefit to also including seasonal mean results, e.g. for DJF and JJA seasons.

--Reply. We agree with this comment. We have rewritten **Section 3.3** and added three related paragraphs for the analysis of seasonal results. Please find further details in **L311-354**.

Line 131: Make clear that you are comparing the CMIP6-PMIP4 PI simulations with observations here. This should also be stated in the caption for Figure 1.

--Reply. We agree with this comment and have revised it.

Line 135: According to the legend, Figure 1c shows the multi-model *median* not the mean – if you are using this to argue that the multi-model mean should be used elsewhere, I suggest modify Figure 1c to show the multi-model mean not median.

--Reply. Thanks for noticing this typo, which we have corrected in the revised manuscript. The figure indeed shows the multi-model mean result, not the multi-model median.

Line 159-161: Note that some of the discussion in Reeves et al. (2013) of changes towards wetter/drier conditions is expressed relative to the early Holocene, not to the late Holocene or pre-industrial. It may also be worth including comparison with Petherick et al. (2013) paper about temperate Australian records:

Petherick, L., Bostock, H., Cohen, T. J., Fitzsimmons, K., Tibby, J., Fletcher, M. S., ... & Dosseto, A. (2013). Climatic records over the past 30 ka from temperate Australia–a synthesis from the Oz-INTIMATE workgroup. Quaternary Science Reviews, 74, 58-77.

--Reply. We agree with this comment and have added Petherick et al. (2013) work in the analysis.

- **L221-227**: For northern Australia, recent studies indicate that the contraction of the local ITCZ contributed to reduced monsoon activity and precipitation (Reeves et al., 2013; Proske et al., 2014; Field et al., 2017; Lowry and McGowan, 2024), which aligns with PMIP4 simulation results (**Figures 1b, 1f**). For temperate Australia, some sites of pollen reconstructions and paleo-hydrologic records of the Oz-INTIMATE series in southeastern Australia indicate wetter conditions with enhanced river discharge and increased precipitation during the mid-Holocene (Petherick et al., 2013; Lowry and McGowan, 2024), while PMIP4 simulations do not consistently capture those robust changes in **Figure 1b**.

Figure 1: (c) Caption should cite Adler et al. 2003 for GPCP. (f) Are different dot sizes significant? Explain in caption if so. I also suggest reversing the colour scheme as red is usually dry and blue is usually wet for precipitation anomaly plots.

--Reply. We agree with this comment and have revised the caption accordingly. Large dots represent significant changes of reconstructed annual precipitation, and small ones are not significant, following the similar plotting method in Bartlein et al. (2011). Additionally, to keep the same color scale meaning in Figures 1b, 1e, and 1f (red colors denote positive anomalies, blue colors denote negative anomalies), we keep the color scheme unadjusted in Figure 1.

Table 1: Table caption should be expanded to define all abbreviations used in table and distinguish between Method I and Method II – remind the reader how these differ (using streamfunction values at 500 hPa versus average over 200-900 hPa).

--Reply. Thanks for the suggestion and we have revised the caption of **Table 1**.

Line 167-168: The methods used to define the ITCZ location need to be briefly described in the Methods section, not just in the Supplement.

--Reply. We agree with this comment and have added a new **Section 2.3** for defining ITCZ position metrics. Please see further details in **L133-141** and Eqs. (3)-(4).

Figure 2: What are the red and blue line colours in (a) and (b) panels? Define in caption.

--Reply. Red is for positive and blue for negative changes. We have revised the caption accordingly.

For panel c, either include full names (inner edge, southern edge etc.) or define the abbreviations in the figure caption: "southern edge of Hadley Cell (Edge_S)" etc.

--Reply. Thanks for the suggestion. We have revised the caption.

Figure 3: Please either label the models on the x-axis of figures or ensure the order of models in the legend matches the order in the plots. Otherwise, any colour-blind reader will not be able to interpret this figure.

--Reply. Thanks for the suggestion. We have revised the order of the legend to be the same as the order on the x-axis.

Line 196: Previous studies of global warming – clarify whether these are model or observational studies and whether focused on historical period or future projections or both.

--Reply. We agree with the comment and have revised it with clarification for future projections. **L259-262**: Previous studies of global warming based on observations and climate simulations reveal that the deep-tropics squeeze and the projected northward migration of ITCZ tend to influence the changes in the width and strength of tropical overturning circulation (Hadley cell) (Kang and Lu, 2012; Lau and Kim, 2015; Byrne et al., 2018; Watt-Meyer and Frierson, 2019; Lionello et al., 2024).

Figure 4 a-f: It is difficult to distinguish the lines based on the colour alone. I suggest using different dot and dash patterns as well as different colours.

--Reply. We agree with the comment and have used different line types and markers to make the identification of the individual lines in this figure easier.

Section 4, paragraphs 2 and 3: In this section, you should also compare your results with D'Agostino et al. (2019) and (2020) studies on NH and SH monsoon changes in PMIP mid-Holocene simulations. These studies make use of MSE budget analysis so they are highly relevant. You may also want to mention these studies in the introduction in the section discussing the MH monsoon changes (lines 35-40).

D'Agostino, R., Bader, J., Bordoni, S., Ferreira, D., & Jungclaus, J. (2019). Northern Hemisphere monsoon response to mid-Holocene orbital forcing and greenhouse gas-induced global warming. *Geophysical Research Letters*, *46*(3), 1591-1601.

D'Agostino, R., Brown, J. R., Moise, A., Nguyen, H., Dias, P. L. S., & Jungclaus, J. (2020). Contrasting southern hemisphere monsoon response: MidHolocene orbital forcing versus future greenhouse gas–induced global warming. *Journal of Climate*, *33*(22), 9595-9613.

--Reply. We agree with this comment and appreciate the suggestion. We have rewritten **Section 3.3** and further compared our results with D'Agostino et al. 2019, 2020, and other related studies. Please see **L311-354** for further details.

---

## Author Comment (AC2)

**Overview:**

This study investigates the effects of the northward migration of the Intertropical Convergence Zone (ITCZ) on the Hadley cell and global hydrological patterns during the mid-Holocene, around 6,000 years ago. This period, known as the "Green Sahara," experienced significant climatic changes, with increased precipitation in typically arid regions. The study utilizes simulations from the PMIP4-CMIP6 archive to analyze shifts in the ITCZ and changes in Hadley cell characteristics, revealing a contraction and weakening of the northern Hadley cell and an expansion and strengthening of the southern cell. These dynamics contributed to wetter conditions in the Northern Hemisphere and drier conditions in the Southern Hemisphere. This study underscores the complex interactions among orbital forcing, ITCZ migration, Hadley cell dynamics, and terrestrial aridity during the mid-Holocene. While climate models provide valuable insights, recognizing their limitations and the uncertainties in simulations and reconstructions is crucial. Continued research and model development are necessary to enhance our understanding of past climate changes and their implications for future climate scenarios. In general, I find the paper is well written and could be published after revision.

--We sincerely thank the reviewer for the thoughtful and constructive comments, which have significantly improved the quality of our manuscript. We have made substantial revisions and greatly extended the manuscript, with four new figures added for the analysis. Please find our point-to-point responses (in blue) to the reviewer comments (in black).

My main concerns:

What are the new findings of the study? The change of Hadley cell, ITCZ, and precipitation patterns in the Mid-Holocene were well reported in previous studies, and especially the Mid-Holocene ITCZ migration and precipitation patterns have been reported in the authors' previous published paper of Bian et al. (2024). Maybe the moisture static energy budget could be interesting to understand the dynamics of changes in Hadley cells. The author should clearly state what is new in the introduction.

Bian, J., Räisänen, J. Mid-Holocene changes in the global ITCZ: meridional structure and land–sea rainfall differences. Clim Dyn 62, 10683–10701 (2024). https://doi.org/10.1007/s00382-024-07470-1

--Reply. Thanks for the suggestion. In the revision, we have made substantial updates to better clarify our study focus and key findings.

- We have added a short paragraph to the end of the Introduction to clarify our main scientific goals in this study. **L91-95**: This work expands over earlier research in Bian and Räisänen (2024) by quantitatively addressing three key aspects: (1) the dynamic connection between the northward ITCZ shift and Hadley cell changes during the mid-Holocene; (2) the joint influence of the ITCZ-Hadley cell evolution on the mid-Holocene hydrological cycle; and (3) proxy evidence alongside physical regimes modulating terrestrial hydroclimate and land aridity during this period.

- As summarized on **L489-498**: The northward migration of ITCZ is accompanied by a northward movement of the inner HC edge, resulting in a contracted and weakened northern HC, while the southern HC expands and intensifies in the mid-Holocene. Specifically, the northern HC width contracted by 1.1° and 0.5°, with strength reductions of 3.7% and 4.1%, while the southern HC expanded by 1.2° and 0.6° and strengthened by 2.9% and 1.8%, according to the two streamfunction metrics.

- Seasonal analysis (**L323-354 and Figs. 5-6**) indicates that the terrestrial hydrological cycle changes are primarily due to summer dynamics with an amplified inter-hemispheric contrast and asymmetry during the mid-Holocene, while there are minor changes during winter seasons for both hemispheres.

- Orbital forcing does not directly drive the hemispheric asymmetry in annual atmospheric radiation balance during the mid-Holocene, and enhanced cloudiness and water vapor in the Northern Hemisphere tropics play key roles. See **L364-377** and **Figs. A1-A2**.

- Moist static energy budget analysis reveals that stronger rising motion significantly promotes vertical MSE advection over land in the Northern Hemisphere, enhancing moist convection and precipitation, while reduced rising motion weakens vertical MSE advection in the Southern Hemisphere, suppressing moist convection and precipitation over land (cf. paragraph on **L385-393**)

- As summarized on **L526-535 and discussed in more detail on L437-467**: The northward ITCZ migration and the associated changes in Hadley cell significantly influence regional climates by altering terrestrial dryness across the tropics and subtropics. Both simulations and reconstructions generally agree on reduced terrestrial

aridity and drylands contraction in the Northern Hemisphere, while the Southern Hemisphere has enhanced aridity and drylands expansion.

Why is the ITCZ migration leading to the change of Hadley Cells? It could also be the Hadley cell leads the ITCZ migration. Where I can see the evidence of ITCZ migration leading to the changes in Hadley Cells?

--Reply. We agree with the comment and acknowledge that this is indeed a very important question. The ITCZ is co-located with the rising branch of Hadley cell, and the northward migration of ITCZ is accompanied by a northward movement of the inner HC edge during the mid-Holocene. This further results in a contracted and weakened northern HC, while the southern HC expands and intensifies. Meanwhile, the changes between ITCZ and Hadley cell are closely linked but their dependency is non-linear and complex (Watt-Meyer and Frierson, 2019), which highlights the intricate nature of the cause-and-effect dynamics between them.

- To better clarify this connection and the complicated cause-effect relationship, we have revised the title to: **Mid-Holocene ITCZ migration: connection with Hadley cell dynamics and impacts on terrestrial hydroclimate**. Furthermore, we have greatly extended the discussion of the ITCZ-Hadley cell dynamics in the **Introduction**, **Section 3.3**, and **Section 4**.

- We have rewritten the Introduction to explain the dynamic connection between ITCZ and Hadley cell and its possible limitations. **L47-61**: Kang et al. (2008) demonstrated an anti-correlation between the annual ITCZ position and the cross-equatorial energy flux (EFE): the northward (southward) migration of the annual ITCZ position corresponds to an anomalous southward (northward) EFE. Further studies indicate that, in the annual mean, the global ITCZ position, the EFE, and the inner edge of global Hadley cell are nearly co-located (Donohoe et al., 2013; Donohoe and Voigt, 2017; Hill, 2019; Kang, 2020; Geen et al., 2020). This co-location implies that the ITCZ and the shared rising branch of the Hadley cell rely on atmospheric energy transport from the hemisphere in which the ITCZ is situated, with the energy export proportional to the ITCZ displacement away from the equator. Consequently, the ITCZ remains aligned with the ascending branch of the Hadley cell, meaning that changes in tropical precipitation with the ITCZ necessitate concurrent changes in Hadley cell (Donohoe and Voigt, 2017; Hill, 2019). Therefore, the ITCZ shift has a correlation with changes

in the edges, width, and intensity of the Hadley cell in the mid-Holocene (Diaz and Bradley, 2004; Lau and Kim, 2015; Donohoe et al., 2013; Donohoe and Voigt, 2017; Byrne et al., 2018; Hill, 2019; Kang, 2020).

- **L73-80**: The energetic constraint theory offers a quantitative framework linking the annual ITCZ position and the Hadley cell, predicting that the annual ITCZ location is near the latitude ($\phi_{EFE}$) at which the vertically integrated meridional MSE flux within the Hadley cell approaches zero (Kang et al., 2008; Adam et al., 2016; Wei and Bordoni, 2018; Kang, 2020; Geen et al., 2020; Lionello et al., 2024). However, this theory also has limitations on shorter time scales and regional ITCZ variations (Roberts et al., 2017; Kang, 2020), suggesting a potential gap between the seasonal evolution of $\phi_{EFE}$ and the ITCZ seasonal and regional migrations (Donohoe and Voigt, 2017; Wei and Bordoni, 2018; Kang et al., 2018; Kang, 2020; Geen et al., 2020). For this study, we primarily focus on annual changes in the global ITCZ position and its possible connection to Hadley cell changes from the PI to the mid-Holocene.

- **L272-280:** It is worth noting that while the changes between the ITCZ and the Hadley cell were closely linked during the mid-Holocene as discussed above, their relationship is inherently nonlinear and complex (Watt-Meyer and Frierson, 2019). Although they share the ascending branch near the Equator and are correlated via the cross-equatorial atmospheric energy flux under annual-mean state (Watt-Meyer and Frierson, 2019; Hill, 2019), a complete understanding of the factors driving their changes may be difficult to achieve when considering the influence from the extratropics and multiple climate drivers (Byrne et al., 2018; Kang et al., 2018; Kang, 2020; Geen et al., 2020; Lionello et al., 2024). To simplify this complexity, the classical energetic theory for the ITCZ position assumes that cross-equatorial atmospheric energy flux arises entirely from the Hadley cell, with negligible contributions from transient and stationary eddies (Byrne et al., 2018; Kang, 2020).

- **L541-557:** The core of energetic theory assumes that the atmospheric cross-equatorial energy flux transport is primarily driven by the zonal mean meridional circulation (i.e. Hadley cell), with contributions from stationary and transient eddy fluxes and changes in gross moist stability (GMS) considered negligible (Adam et al., 2016; Donohoe and Voigt, 2017; Wei and Bordoni, 2018; Hill, 2019; Kang, 2020). However, both observations and climate simulations reveal that eddy fluxes play a non-negligible role in energy and moisture budgets in the tropical atmosphere (Roberts et al., 2017; Byrne et al., 2018; Geen et al., 2020; Kang, 2020). This means energetic constraint theories

linking the ITCZ and Hadley cell have limitations in explaining ITCZ changes and their nonlinear dependency on the Hadley cell changes. For instance, the Hadley cell did not exert a dominant influence on changes in atmospheric heat transport across the Equator during glacial periods when tropical rainfall and meridional heat transport underwent substantial changes (Donohoe et al., 2013, 2014; McGee et al., 2014; Roberts et al., 2017). Roberts et al. (2017) highlights that there is insufficient evidence to support the Hadley cell as the primary driver of tropical rainfall and energy transport, particularly when considering the effect of stationary and transient energy fluxes. Moreover, energetic theory for the zonal-mean framework cannot fully account for the relationship between regional rainfall changes and meridional migrations of the ITCZ, as the direction and magnitude of regional rainfall band migration vary significantly by longitude (Roberts et al., 2017; Atwood et al., 2020). Additionally, even when Hadley cell meridional circulation dominates cross-equatorial atmospheric energy transport, the energetic framework may still be insufficient, as changes in GMS can also make a contribution (Wei and Bordoni, 2018; Kang, 2020).

The asymmetry responses of Hadley Cells in the Northern Hemisphere and Southern Hemisphere are interesting. It would be great if the authors could show a more dynamic understanding of the asymmetry responses, especially more focus on the moisture static energy budget in the revision.

--Reply. We agree that the asymmetric response of Hadley cells is a very interesting and important question.

- **Figures 4c, 4d, and 4f** illustrate that the increased terrestrial rainfall is closely linked to positive anomalies in atmospheric energy divergence ($\nabla \cdot Fa > 0$) over the Northern Hemisphere. This is primarily driven by enhanced vertical moist static energy (MSE) advection over land (Terms III and IV), while changes in horizontal MSE advection (Terms I and II) and eddy MSE flux (Term V) have a compensating influence, as illustrated in **Figure 4f**.

- For the $\delta \nabla \cdot Fa > 0$ and associated changes in MSE vertical advection over land in the Northern Hemisphere: Changes in radiative forcing due to orbital parameters result in increased net radiation ($\delta Ra > 0$) in the Northern Hemisphere during spring and summer (Brierley et al., 2020). This alters the net energy flux divergence ($\delta \nabla \cdot Fa$), which further

influences the temperature and moisture distributions and affects the MSE gradient and advection over land during the mid-Holocene (**Figures 4b, 4d**), altering the energy balance of the atmosphere ($\delta \nabla \cdot Fa$). This further influences the temperature and moisture distributions and affects the MSE gradient and advection over land. See also **L359-364** in the manuscript.

- ■ However, in the annual mean, orbital forcing is symmetric between the two hemispheres, while the change in the annual mean atmospheric radiation balance is asymmetric, with a slight positive anomaly ($\delta Ra > 0$) from $10°N$ to $30°N$ for land and ocean (**Figure 4b**), and $15°N$ to $40°N$ for land alone (**Figure 4d**). This suggests that factors other than the direct orbital forcing are also important. See also **L364-367** in the manuscript.

- ■ Analysis of the separate contributions of the SW and LW flux components to the atmospheric radiation balance shows reduced NH surface shortwave radiation alongside increased atmospheric shortwave absorption, indicating that enhanced cloudiness and water vapor play key roles (**Figure A1**).

- ■ Changes in the annual net surface energy budget ($\delta(Rs - SH0 - LE)$) over the ocean are slightly positive in the Southern Hemisphere tropics but negative in the Northern Hemisphere tropics (**Figure A2**). This suggests that ocean currents are transferring more heat from the Southern Hemisphere to the Northern Hemisphere, thereby amplifying the inter-hemispheric asymmetry in the ocean-to-atmosphere energy transfer and annual atmospheric radiation balance during the mid-Holocene. See also **L378-383** in the manuscript.

In most of the analysis, this study uses multiple model ensemble mean, I am wondering whether there are any models and realizations that could better capture the reconstructed precipitation patterns, or how different the simulated precipitation patterns change among different models. For example, does a model with a larger Hadley Cell northward shifting and intensifying show more precipitation increases in the Northern Hemisphere extra-tropical monsoon regions? I think some more discussions on the model uncertainties might further help to understand the dynamic links between the large-scale circulation change and precipitation patterns.

--Reply. We agree with the comment and acknowledge that although all nine simulations show a northward migration and zonal precipitation belt during the mid-Holocene (not shown; Bian

and Räisänen, 2024), intermodel differences in precipitation changes may introduce uncertainties when these results are compared with paleoclimate reconstructions (McGee et al., 2014; Byrne et al., 2018; Bian and Räisänen, 2024). The metrics used to quantify Hadley cell width and strength can significantly affect the patterns and magnitude of mid-Holocene changes.

Among the nine models evaluated, MRI-ESM2-0, MPI-ESM1-2-LR, and EC-EARTH3-LR exhibit larger shifts in Hadley cell edges and width, alongside greater precipitation increases in the Northern Hemisphere extra-tropics (**Table 1, Figure 3**). These three models also demonstrate relatively better performance in capturing ITCZ changes, based on multiple metrics noted by Bian and Räisänen (2024). However, we chose to not discuss the behaviour of the individual models in more detail, to avoid a further lengthening of the manuscript.

**References**

Adam, O., Bischoff, T., and Schneider, T.: Seasonal and interannual variations of the energy flux equator and ITCZ. Part I: Zonally averaged ITCZ position, Journal of Climate, 29, 3219–3230, 2016.

Atwood, A. R., Donohoe, A., Battisti, D. S., Liu, X., and Pausata, F. S.: Robust longitudinally variable responses of the ITCZ to a myriad of climate forcings, Geophysical Research Letters, 47, e2020GL088 833, 2020.

Back, L. and Bretherton, C.: Geographic variability in the export of moist static energy and vertical motion profiles in the tropical Pacific, Geophysical research letters, 33, 2006.

Bordoni, S. and Schneider, T.: Monsoons as eddy-mediated regime transitions of the tropical overturning circulation, Nature Geoscience, 1, 515–519, 2008.

Brierley, C. M., Zhao, A., Harrison, S. P., Braconnot, P.,Williams, C. J., Thornalley, D. J., Shi, X., Peterschmitt, J.-Y., Ohgaito, R., Kaufman, D. S., et al.: Large-scale features and evaluation of the PMIP4-CMIP6 mid-Holocene simulations, Climate of the Past, 16, 1847–1872, 2020.

Byrne, M. P., Pendergrass, A. G., Rapp, A. D., and Wodzicki, K. R.: Response of the intertropical convergence zone to climate change: Location, width, and strength, Current Climate Change Reports, 4, 355–370, 2018.

Claussen, M., Dallmeyer, A., and Bader, J.: Theory and modeling of the African humid period and the green Sahara, in: Oxford research encyclopedia of climate science, Oxford University Press, 2017.

Diaz, H. F. and Bradley, R. S.: The Hadley circulation: Present, past, and future: An introduction, in: The Hadley circulation: present, past and future, Springer, 2004.

Donohoe, A. and Voigt, A.: Why future shifts in tropical precipitation will likely be small: The location of the tropical rain belt and the hemispheric contrast of energy input to the atmosphere, Climate Extremes: Patterns and Mechanisms, pp. 115–137, 2017.

Donohoe, A., Marshall, J., Ferreira, D., and Mcgee, D.: The relationship between ITCZ location and cross-equatorial atmospheric heat transport: From the seasonal cycle to the Last Glacial Maximum, Journal of Climate, 26, 3597–3618, 2013.

Geen, R., Bordoni, S., Battisti, D. S., and Hui, K.: Monsoons, ITCZs, and the concept of the global monsoon, Reviews of Geophysics, 58, e2020RG000 700, 2020.

Harrison, S., Bartlein, P., Brewer, S., Prentice, I., Boyd, M., Hessler, I., Holmgren, K., Izumi, K., and Willis, K.: Climate model benchmarking with glacial and mid-Holocene climates, Climate Dynamics, 43, 671–688, 2014.

Hill, S. A.: Theories for past and future monsoon rainfall changes, Current Climate Change Reports, 5, 160–171, 2019.

Hill, S. A., Ming, Y., Held, I. M., and Zhao, M.: A moist static energy budget–based analysis of the Sahel rainfall response to uniform oceanic warming, Journal of Climate, 30, 5637–5660, 2017.

Kang, S. M.: Extratropical influence on the tropical rainfall distribution, Current Climate Change Reports, 6, 24–36, 2020.

Kang, S. M., Held, I. M., Frierson, D. M., and Zhao, M.: The response of the ITCZ to extratropical thermal forcing: Idealized slab-ocean experiments with a GCM, Journal of Climate, 21, 3521–3532, 2008.

Lau, W. K. and Kim, K.-M.: Robust Hadley circulation changes and increasing global dryness due to CO2 warming from CMIP5 model projections, Proceedings of the National Academy of Sciences, 112, 3630–3635, 2015.

Lionello, P., D'Agostino, R., Ferreira, D., Nguyen, H., and Singh, M. S.: The Hadley circulation in a changing climate, Annals of the New York Academy of Sciences, 1534, 69–93, 2024.

McGee, D., Donohoe, A., Marshall, J., and Ferreira, D.: Changes in ITCZ location and cross-equatorial heat transport at the Last Glacial Maximum, Heinrich Stadial 1, and the mid-Holocene, Earth and Planetary Science Letters, 390, 69–79, 2014.

Roberts, W. H., Valdes, P. J., and Singarayer, J.: Can energy fluxes be used to interpret glacial/interglacial precipitation changes in the tropics ?, Geophysical Research Letters, 44, 6373–6382, 2017.

Wei, H.-H. and Bordoni, S.: Energetic constraints on the ITCZ position in idealized simulations with a seasonal cycle, Journal of Advances in Modeling Earth Systems, 10, 1708–1725, 2018

---

## Author Response (AR2)

Review of "Mid-Holocene ITCZ migration: Connection with Hadley cell dynamics and impacts on terrestrial hydroclimate"

By Bian et al.

I have previously reviewed the manuscript, and the authors have conducted substantial analysis to address the reviewers' concerns and enhance the study's novelty. They have largely revised the manuscript, making the storyline clearer and easier to follow. I find the paper interesting and valuable for publication. However, it still requires some revisions, as some of our concerns have not been fully addressed.

--We greatly appreciate the constructive comments and kind suggestions from the reviewer, which have significantly enhanced the quality of our manuscript. Please find our responses (in blue) to the reviewer's comments (in black).

1. A clear knowledge gap needs to be clearly defined at the beginning of the introduction. The current introduction is not efficient at this point. I must say I only capture such a piece of information in the main context of lines 275-279. It would be very helpful if the author could write one or two sentences to address the question "What is new for the present study?".

--Reply. Thanks for the suggestion. We have added related contents of knowledge gaps and creative points for this study to the second last paragraph of **Introduction**.

- ■ **L85-93 in the revision with tracked changes**. Despite the many earlier studies, the complex interactions between the ITCZ, Hadley cell and hydrological changes during the mid-Holocene remain insufficiently understood. Therefore, a detailed quantitative evaluation of their dynamic interplay and multi-scale atmospheric processes involved is important, including changes in cross-equatorial energy flux transport by the stationary and transient eddies. This study addresses these gaps by focusing on three key aspects: (1) the dynamic connection between the northward ITCZ shift and Hadley cell changes during the mid-Holocene; (2) the joint influence of the ITCZ-Hadley cell evolution on the mid-Holocene hydrological cycle; and (3) the evaluation of consistency between proxy data and model simulations regarding terrestrial hydroclimate and land aridity in the mid-Holocene. Building on the recent advances in understanding the mid-Holocene global ITCZ by Bian and Räisänen (2024), this work specifically targets these critical research aspects.

2. In the abstract also in the conclusion, the authors have a strong statement of "Orbital forcing during the mid-Holocene does not directly drive the hemispheric asymmetry in annual atmospheric radiation balance." Why? The orbital forcing initialed all the changes in the solar radiation, and thereafter, drove changes in atmospheric circulation and further enhanced the hemispheric asymmetry in annual atmospheric radiation balance. Could you please make it more specific? What do you mean by "directly drive the hemispheric asymmetry in annual atmospheric radiation balance"? Maybe I just missed some key figures or results in the manuscript.

--Reply. To eliminate the risk of misunderstanding, we have rewritten the **L17-18** in the **Abstract** as: Although orbital forcing during the mid-Holocene was symmetric around the equator in the annual mean, it indirectly drove hemispherically asymmetric changes in annual atmospheric radiation balance.

- During the last round's revision, we have added three paragraphs in **Section 3.3** and two extra figures (**Figures A1, A2**) in the **Appendix A** to further clarify the changes in atmospheric radiation balance during the mid-Holocene.

- In the annual mean, orbital forcing is symmetric between the two hemispheres, while the change in the annual mean atmospheric radiation balance is asymmetric, with a slight positive anomaly ($\delta Ra > 0$) from 10°N to 30°N for land and ocean (**Figure 4b**), and 15°N to 40°N for land alone (**Figure 4d**). This suggests that factors other than the direct orbital forcing are also important.

- We further analyze the separate contributions of the shortwave (SW) and longwave (LW) flux components to the atmospheric radiation balance. Please see further details on **L384-393**.

- We further conducted analysis of the annual net surface energy budget ($\delta(Rs - SH0 - LE)$). Please see further details on **L394-401.**

[Figure]

Figure A1. Individual contributions of the shortwave (a) and longwave (b) flux components to the changes of annual atmospheric radiation balance from the PI to the MH period. (c), (d) are as (a), (b) respectively, but for land only. The unit is W m⁻². All results are based onPMIP4-CMIP6 multimodel averages.

[Figure]

Figure A2. Changes in the annual net surface energy budget ($\delta(Rs - SH0 - LE)$) from the PI to the MH period (unit: W m⁻²). All results are based on PMIP4-CMIP6 multimodel averages.

3. The discussion in the conclusion section of lines 538-556 is very interesting and important. I take the conclusion that "the contributions from transient and stationary eddies are not negligible in mid-Holocene", as a key new finding of the present study. However, the discussion here only provides reviews on the debate on the importance of eddy fluxes from previous studies. I would suggest the authors add some findings from your results. What do we learn from your analysis in the mid-Holocene? You have done a comprehensive dynamic analysis to draw a conclusion here.

--Reply. Thanks for the suggestion. We agree that contributions from transient and stationary eddies are not negligible in the mid-Holocene, as shown by both multiple previous studies and our own comparison results. We have added analysis of this in **Section 3.2** and **Appendix B**.

- ■ Our results identified that the annual cross-equatorial energy flux exhibits negative anomalies ($\delta F < 0$) in the deep tropics as well as the Northern Hemisphere, indicating reduced northward energy transport from the Southern Hemisphere during the mid-Holocene. This imbalance shifts the energy flux zero-crossing latitude northward, resulting in a corresponding northward migration of the ITCZ (**Eq. B1** and **Figure B1**). However, as shown in **Eqs. B2-B3** and **Figure B1**, the contributions from the SE and TE eddies changes are not negligible during this period. These complexities highlight the limitations of the energetic constraint framework in fully explaining ITCZ shifts and their nonlinear relationship with Hadley cell dynamics during the mid-Holocene. Please see further details on **L280-296** and **L556-575**.

[Figure]

Figure B1. Changes in the annual atmospheric meridional energy transport by the mean meridional circulation (MMC), stationary eddies (SE), and transient eddies (TE) from the PI to the MH period (unit: PW). See Eqs. B2-B3 in Appendix B for further details. All results are based on PMIP4-CMIP6 multimodel averages.

4. Figure 4 requires efforts to improve, and the same for Figure 6. There are busy figures. The colors and marks make the figures difficult to read. In the legend, the change in precipitation is solid red, but it is dash red in the figure, and the change in P-E is solid blue, but it is dash blue in the figure. My suggestions would be to only use different colors for the terms, darker and thick lines for the key information of the figure, and lighter colors for the less important information.

--Reply. We agree with this comment and have revised them accordingly.

---

## Author Response (AR3)

Dear Editor,

Thanks for the great support and kind suggestions on our submission. Please find our responses (in blue) to the final comments (in black).

L67: and significantly increased monsoonal rainfall

--Revised.

L83: 'between the PI and mid-Holocene'

--Revised.

Section 2.3.: make sure to clearly define all the terms of the different equations, and double check your equations. For example L and q are not defined in the MSE equation, and phi should be replaced by g.z (gravitational acceleration times geopotential height), as phi is already defined in the previous equations as latitude.

--Reply. Thanks for the suggestion and we have revised those places accordingly.

L561-562: "that stationary and transient eddies changes significantly influence on cross-equatorial energy flux transport"

--Revised.

L564-565: This is a strong statement. Unless you really think that the evidence shown in the cited studies is unequivocal, I would suggest toning it down.

--Reply. We agree with the comment and have removed this less relevant sentence.

Table S1: Please also provide references for the mid-Holocene simulations, ie either the manuscripts describing each of the MH simulations or the DOI of each of the MH simulation. Please also move this table as Appendix as these simulations are the basis of your manuscript.

--Reply. Thanks for the suggestion. We have moved it to Appendix A (Table A1) and added one table with DOI details (Table A2).